# DiffCrossGait: Trajectory-Level Alignment for 2D-3D Cross-Modal Gait Recognition via Latent Diffusion

**Zhiyang Lu** [1 2]   **Ming Cheng** [1 2]

## Abstract

Cross-modal 2D–3D gait recognition is impeded by inherent domain discrepancies between 2D silhouette and 3D LiDAR range-view representations. While prior methods align only final embeddings, we propose **DiffCrossGait**, which reformulates cross-modal matching as trajectory-level alignment in an identity-relevant latent diffusion space, rather than assuming full equivalence between 2D and 3D observations. By driving both modalities with shared Gaussian noise within a latent space, we enable continuous alignment throughout the generative evolution. We introduce a **Tri-Phase Alignment Strategy** that exploits varying noise intensities to enforce identity anchoring, dynamics consistency, and cross-modal structural recoverability, thereby constraining both modalities to share denoising dynamics and bottleneck structure, which promotes modality-invariant gait features. Crucially, our framework decouples generative alignment from the discriminative backbone; the diffusion mechanism serves exclusively as a training objective, ensuring high inference efficiency by eliminating the computational overhead of iterative denoising. Extensive experiments on the SUSTech1K and FreeGait benchmarks demonstrate that DiffCrossGait achieves state-of-the-art performance.

## 1. Introduction

Gait is a remote biometric that supports identification at a distance and without contact, making it attractive for surveillance and public-security applications where face or finger-

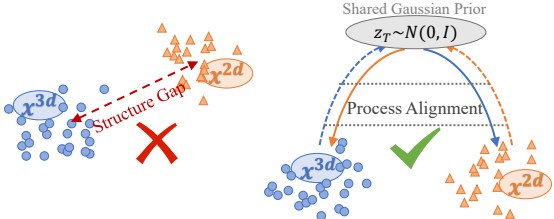

(a) Discriminative Alignment    (b) Our Dynamic Process Alignment

*Figure 1.* **Concept of Dynamic Process Alignment.** **(a)** Static alignment fails to bridge the structural gap between 2D and 3D gait data. **(b)** Our method introduces a Shared Gaussian Prior ($z_T$) to unify the latent space. By constraining the diffusion forward and reverse trajectories to remain synchronized, we achieve robust cross-modal alignment that is superior to matching endpoints.

print acquisition is unreliable (Chen et al., 2024; Li & Zhao, 2022; Fan et al., 2025; Shen et al., 2024; Sarkar et al., 2005; Yang et al., 2025). With the increasing availability of multi-modal sensing, practical systems often observe the same person using both 2D cameras (yielding silhouette sequences) and 3D depth sensors such as LiDAR (yielding geometry-driven sequences). This motivates **cross-modal 2D–3D gait recognition**, where the goal is to retrieve a person's identity across modalities (Guo et al., 2025; Wang et al., 2024; Filipi Gonçalves dos Santos et al., 2022). A central difficulty is that 2D and 3D gait data encode fundamentally different information: silhouettes preserve projected shape contours, whereas LiDAR-derived representations capture spatial geometry. In this work, the alignment is imposed on standard gait-centric representations: 2D silhouettes and LiDAR range views. The latter preserves sensor-native geometric cues in a regular format, whereas prematurely projecting it into pseudo-2D silhouettes may reduce the apparent modality gap at the cost of discarding discriminative depth structure. Therefore, our objective is not to make the two modalities identical at the input level, but to align their identity-relevant latent dynamics. As a result, the two modalities follow distinct distributions and exhibit structural discrepancies in their latent manifolds, especially under covariates such as view change, clothing, carried objects, occlusion, and illumination (Zhang et al., 2024; 2025).

Most existing cross-modal gait methods adopt a discriminative alignment perspective: they train encoders with metric

---
[1]Fujian Key Laboratory of Urban Intelligent Sensing and Computing, Xiamen University, 361005, P.R. China. [2]Key Laboratory of Multimedia Trusted Perception and Efficient Computing, Ministry of Education of China, Xiamen University, 361005, P.R. China. Correspondence to: Ming Cheng <chm99@xmu.edu.cn>.

*Proceedings of the 43rd International Conference on Machine Learning*, Seoul, South Korea. PMLR 306, 2026. Copyright 2026 by the author(s).

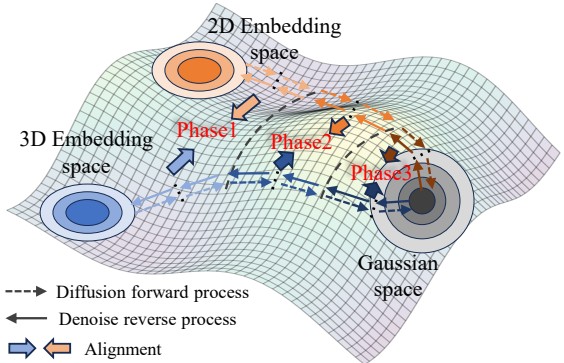

*Figure 2.* **Trajectory alignment across three noise regimes.** We leverage the varying noise levels to impose hierarchical constraints: anchoring identity semantics at high signal retention (Phase 1), synchronizing denoising dynamics at medium levels (Phase 2), and enforcing structural manifold overlap under strong noise (Phase 3). This enables explicit extraction of modal-invariant structures throughout the generative evolution.

learning losses that force 2D and 3D samples of the same identity to be close in a shared embedding space (Jiang et al., 2025; Wang et al., 2024; Yu et al., 2025b; Li et al., 2025). While effective to some extent, this alignment is typically imposed only at the final embedding layer, as shown in Fig. 1. In practice, endpoint-only constraints can be brittle: two modalities may be coerced to match at the output while still following incompatible latent trajectories, causing alignment to break under distribution shifts. Moreover, rigid alignment losses can conflict with discriminative objectives, making optimization unstable when modalities are highly heterogeneous.

This raises a question: **can cross-modal alignment be formulated as a trajectory-level constraint rather than an endpoint constraint?** Diffusion models provide a natural mechanism for trajectory modeling: they progressively map data to a near-Gaussian state via forward noise injection and learn to reverse this process via denoising (Song et al., 2021a; Rombach et al., 2022). Inspired by this property, we propose to embed cross-modal alignment into the diffusion evolution itself by enforcing that the two modalities follow a synchronized diffusion trajectory. We introduce **DiffCross-Gait**, a cross-modal 2D–3D gait recognition framework that performs dynamic process alignment through a unified latent diffusion objective. Starting from sequence-level representations extracted by modality-specific backbones, we inject forward noise using an identical noise schedule and, crucially, share the same Gaussian noise sample at each timestep across modalities. A lightweight, parameter-shared denoiser then learns to predict the shared noise and to produce an intermediate bottleneck representation, enabling alignment constraints to act throughout the latent evolution.

To exploit the fact that different noise levels correspond

to different semantic granularity, DiffCrossGait applies a **Tri-Phase Alignment Strategy** along the diffusion timeline, as shown in Fig. 2: Phase1: at small noise, we anchor identity semantics to prevent drift; Phase2: at medium noise, we align denoising dynamics and bottleneck states to synchronize evolution; and Phase3: at large noise, we enforce cross-modal structural recoverability via cross-conditioning, explicitly narrowing the modality gap when signal is weak. Importantly, diffusion is used only during training as an auxiliary objective: at inference time, we discard the diffusion branch and perform recognition using the discriminative backbone, avoiding any iterative denoising overhead. The contributions are summarized as follows:

- We propose DiffCrossGait, a diffusion-regularized framework for cross-modal 2D–3D gait recognition that upgrades alignment from endpoint matching to trajectory-level process alignment via shared-noise latent diffusion.

- We introduce a Tri-Phase Alignment Strategy that applies identity anchoring, dynamics consistency, and cross-modal structural recoverability across noise regimes, yielding stronger modality-invariant representations.

- We design multi-level consistency objectives (noise prediction, bottleneck bridging state, and reconstruction) and show that diffusion can serve as a training-only objective, preserving inference efficiency.

- Extensive experiments on standard benchmarks, including SUSTech1K and FreeGait, demonstrate that DiffCrossGait achieves state-of-the-art (SOTA) performance.

**Conflict of Interest Disclosure.** The authors declare no financial conflicts of interest that could reasonably be perceived to influence this work.

## 2. Related Work

### 2.1. Cross-modal Gait Recognition

The evolution of gait recognition has transitioned from single-modal analysis (typically 2D silhouettes) to complex cross-modal settings (Fu et al., 2023; Jin et al., 2025; Huang et al., 2021b; Chao et al., 2019; Fan et al., 2020; Han & Bhanu, 2005; Yang et al., 2025; Chao et al., 2021; Wang et al., 2025a; Huang et al., 2021a; Ma et al., 2024; Filipi Gonçalves dos Santos et al., 2022; Wang et al., 2025b; Huang et al., 2025; Habib et al., 2025), necessitated by the proliferation of 3D sensors (Han et al., 2024; Shen et al., 2025; Lu et al., 2025; Zheng et al., 2022; Shen et al., 2023). Early approaches in this domain, such as GaitSet (Chao et al., 2021) and its derivatives, focused exclusively on 2D silhouette sequences, learning temporal representations through set-based pooling. However, the advent of 2D-3D

cross-modal tasks introduced significant challenges due to the inherent heterogeneity between projected shape contours and spatial geometric structures (Zhang et al., 2024; 2025; Zuo et al., 2024). Recent advancements have attempted to bridge this modality gap primarily through metric learning and subspace alignment. LidarGait (Shen et al., 2023) pioneered the benchmarking of LiDAR-based gait but relied on projecting 3D data into 2D depth maps, partly discarding geometric fidelity. Subsequent works like CrossGait (Wang et al., 2024) and CL-Gait (Guo et al., 2025) introduced more sophisticated feature mapping techniques to align representations in a shared embedding space. Beyond gait, the challenge of aligning heterogeneous modalities is extensively studied in Visible-Infrared Person Re-identification (VI-ReID). IDKL (Ren & Zhang, 2024) and TVI-LFM (Hu et al., 2024) employ implicit discriminative knowledge learning or leverage large foundation models to extract modality-invariant features. Other recent approaches like TSKD (Shi et al., 2026) utilize two-stage knowledge distillation to transfer semantic cues, while SCR (Yu et al., 2025a) explores suggestive clues guidance to refine cross-modal matching. While these methods achieve reasonable performance, they fundamentally rely on static endpoint alignment—forcing alignment and fail to model the structural evolution of features within the latent space, often resulting in superficial alignment that collapses under significant covariant shifts. In contrast, our framework shifts the paradigm from static constraint to dynamic process alignment, ensuring consistency throughout the feature evolution trajectory.

## 2.2. Diffusion for Cross-Modal Alignment.

Recently, diffusion models have transcended their generative roots to serve as powerful representation learners (Rombach et al., 2022). In the realm of cross-modal person re-identification, methods like DiVE (Dai et al., 2025) and IA-Diff (Yu et al., 2025c) employ diffusion to generate synthetic cross-modal samples, treating the domain gap as a data scarcity problem. However, these pixel-level generation approaches incur high computational costs and do not explicitly align the latent manifolds of real data. On the other hand, ContextDiff (Yang et al., 2024) and Align-Diff (Qiu et al., 2024) demonstrate that modulating the diffusion trajectory itself can enforce semantic alignment between heterogeneous inputs. We advance this direction by introducing a unified diffusion trajectory for 2D-3D gait, using the shared denoising process as a dynamic structural constraint for the discriminative backbone.

## 3. Methodology

### 3.1. Feature Extraction

Given input sequences from distinct modalities, denoted as $x^{2d}, x^{3d} \in \mathbb{R}^{B \times S \times C \times H \times W}$, we first employ modality-specific backbones to extract spatiotemporal features. Subsequently, a Temporal Pooling (TP) layer (Fan et al., 2025) aggregates these features along the temporal dimension to obtain sequence-level feature maps:

$$h^m = \text{TP}\left(\text{Backbone}_m\left(x^m\right)\right) \in \mathbb{R}^{B \times C \times H \times W}, \quad (1)$$

where $m \in \{2d, 3d\}$. To reduce interference between recognition and diffusion objectives, we map $h^m$ through two separate 1×1 heads: a discriminative head for retrieval $z^m_{disc}$ and a generative head for diffusion regularization $z^m_{gen}$, as in Fig. 3. In the discriminative branch, we apply Horizontal Pyramid Pooling (HPP) (Chao et al., 2021; Fan et al., 2023) along the spatial dimension to perform multi-scale spatial partitioning, yielding part-level features:

$$g^m_{disc} = \text{HPP}\left(z^m_{disc}\right) \in \mathbb{R}^{B \times C \times P}. \quad (2)$$

Subsequently, these features are projected into a unified embedding space via Shared Fully Connected (FC) layers:

$$e^m_{disc} = \text{SharedFCs}\left(g^m_{disc}\right) \in \mathbb{R}^{B \times D \times P}. \quad (3)$$

Finally, we employ a Shared BNNeck module (Fan et al., 2025) to generate the class logits required for classification:

$$l^m_{disc} = \text{SharedBNNecks}\left(e^m_{disc}\right) \in \mathbb{R}^{B \times K \times P}, \quad (4)$$

where $D$ denotes the dimension of the shared embedding, $K$ represents the number of training identities, and $P$ indicates the number of spatial parts. Consistent with established protocols (Wang et al., 2024; Guo et al., 2025), $e^m_{disc}$ is employed to compute the cross-modal triplet loss, while $l^m_{disc}$ is supervised via cross-entropy loss:

$$\mathcal{L}_{\text{disc}} = \mathcal{L}_{\text{tri}}\left(e^{2d}_{disc}, e^{3d}_{disc}\right) + \mathcal{L}_{\text{ce}}\left(l^{2d}_{disc}, y\right) + \mathcal{L}_{\text{ce}}\left(l^{3d}_{disc}, y\right), \quad (5)$$

where $\mathcal{L}_{tri}$ enforces cross-modal alignment via bidirectional triplet sampling, and $y$ denotes the ground truth identity labels. In parallel, the generative projection $z^m_{gen}$ functions directly as the clean latent state for the Latent Diffusion Model (LDM), establishing the ground truth for the diffusion trajectory and reconstruction objectives. Concurrently, we apply Global Average Pooling (GAP) along the spatial dimension of $z^m_{gen}$ to extract an identity-semantic prior:

$$c^m_{id} = \text{GAP}(z^m_{gen}) \in \mathbb{R}^{B \times C}, \quad (6)$$

which serves as the high-level condition for the subsequent denoising U-Net. This decoupled-head architecture not only safeguards the integrity and stability of the discriminative branch but also isolates a dedicated latent pathway for the diffusion module, thereby mitigating task interference between discriminative learning and generative modeling. This decoupling also specifies the scope of our shared-dynamics assumption: paired 2D-3D samples are

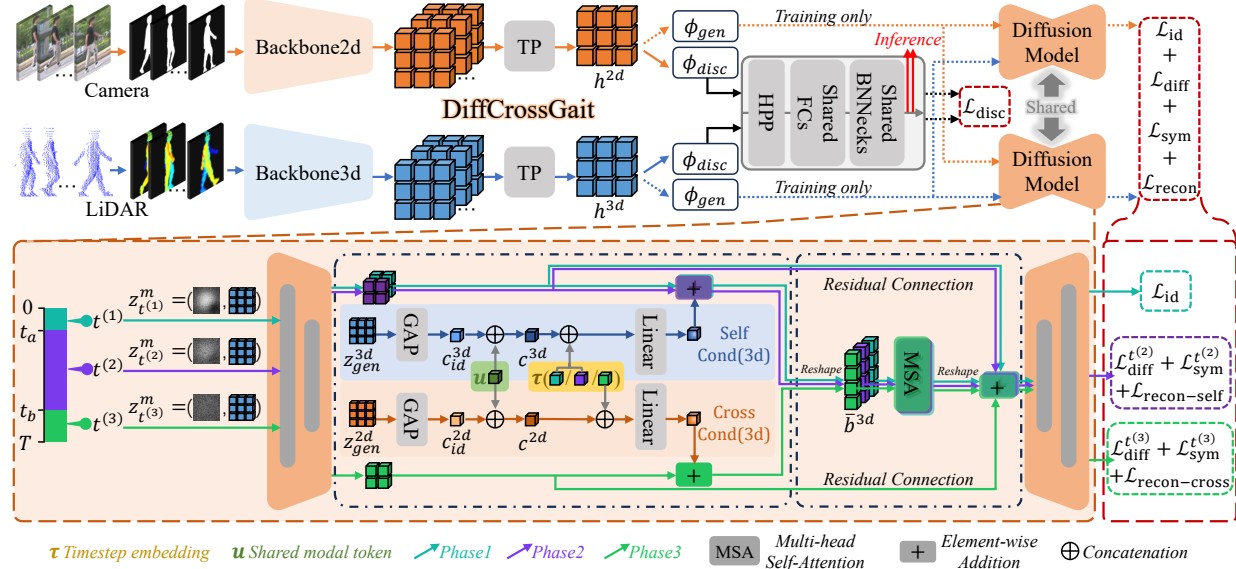

*Figure 3.* **Overview of the DiffCrossGait framework.** The architecture consists of two parallel streams: **(Top) The Discriminative Backbone** extracts modality-specific features from 2D video and 3D point cloud sequence, disentangling them into discriminative and generative components. Note that *only the discriminative branch is retained for efficient inference.* **(Bottom) The Auxiliary Diffusion Branch (Training Only)** aligns the modalities via a shared generative process.

encouraged to share compatible denoising dynamics only in the identity-relevant generative subspace, while modality-specific cues can remain outside this regularized pathway. Thus, diffusion serves as a training-time structural regularizer rather than a constraint of full modality equivalence.

### 3.2. Unified Diffusion Trajectory

We formulate the latent diffusion as a time-dependent discrete stochastic process. Given $T$ diffusion steps and a noise schedule $\{\beta_t\}_{t=1}^T$, the signal retention coefficients are defined as $\alpha_t = 1 - \beta_t$ and $\bar{\alpha}_t = \prod_{s=1}^t \alpha_s$. By sampling a timestep $t \sim \mathcal{U}_{1,\dots,T}$ and Gaussian noise $\boldsymbol{\epsilon} \sim \mathcal{N}(0, \mathbf{I})$, the cross-modal forward diffusion process is defined as:

$$z_t^m = \sqrt{\bar{\alpha}_t} z_0^m + \sqrt{1 - \bar{\alpha}_t} \boldsymbol{\epsilon}, \qquad (7)$$

where $z_0^m$ corresponds to the clean sequence-level latent (i.e., $z_{gen}^m$). Crucially, paired 2D and 3D latents use the same noise realization $\boldsymbol{\epsilon}$. This shared stochastic driver does not force identical raw geometry; instead, it makes the denoiser learn compatible score/denoising fields for the identity-relevant components of both modalities. The resulting synchronization provides a trajectory-level regularization signal beyond endpoint matching. For the reverse process, we employ a parameter-shared lightweight denoising network $U_\theta$ to predict the noise and extract a bottleneck bridging state simultaneously:

$$(\hat{\boldsymbol{\epsilon}}_t^m, mid_t^m) = U_\theta(z_t^m, \tau(t), c). \qquad (8)$$

Here, $\tau(t)$ denotes the timestep embedding, and $c = [u, c_{id}^m] \in \mathbb{R}^{d_{mod}+C}$ represents the high-level condition

formed by concatenating a learnable shared modal token $u \in \mathbb{R}^{d_{mod}}$ with the modality-specific identity prior. We design two routing mechanisms for this condition: **1) Self-modal conditioning (`SelfCond`):** The denoiser is conditioned on the modality's own prior (i.e., $c^{2d}$ drives $z_t^{2d}$). **2) Cross-modal conditioning (`CrossCond`):** The identity semantics are swapped, such that the 2D denoising process is driven by $c^{3d}$, and vice versa. The term $mid_t^m$ represents the bottleneck bridging state within the U-Net. As the central structural bottleneck through which the denoising flow must propagate, it offers a more direct constraint on the evolutionary consistency of the two modalities compared to endpoint embeddings alone.

**Lightweight Denoising Architecture.** Unlike deep multi-scale U-Net designed for high-fidelity image generation, our network $U_\theta$ is constructed as a lightweight module optimized specifically for cross-modal representation learning. To capture global dependencies with minimal computational overhead, we introduce an axial attention mechanism within the bottleneck stage. For the bottleneck feature $b^m \in \mathbb{R}^{B \times C' \times H' \times W'}$, we first concatenate the timestep embedding and condition vector, projecting them to generate a channel-wise bias:

$$\Delta^m = W[\tau(t), c] \in \mathbb{R}^{B \times C' \times 1 \times 1}. \qquad (9)$$

This bias is injected into the feature map via broadcasting:

$$\tilde{b}^m = b^m + \Delta^m. \qquad (10)$$

Subsequently, we unfold the feature map solely along the vertical spatial axis to construct a sequence set

**Algorithm 1** Training Pipeline of DiffCrossGait

---

1: **Input:** Data $(x^{2d}, x^{3d}, y)$; Steps $T$; Thresholds $\rho_a, \rho_b$; Weights $\{\lambda\}$.
2: **Output:** Optimized parameters $\Theta$.
3: Initialize schedule $\bar{\alpha}_t$ and phase boundaries $t_a, t_b$ based on thresholds.
4: **repeat**
5:     Sample batch data $(x^{2d}, x^{3d}, y)$, shared noise $\epsilon \sim \mathcal{N}(0, \mathbf{I})$.
6:     Extract decoupled features$(z_{disc}^m, z_{gen}^m)$ and logits; Initialize loss $\mathcal{L} = \{\lambda_1 \mathcal{L}_{\text{disc}}\}$.
7:     Sample tri-phase time steps $t^{(1)} \sim [1, t_a]$, $t^{(2)} \sim (t_a, t_b]$, $t^{(3)} \sim (t_b, T]$.
8:     **for** $k = 1$ **to** 3 **do**
9:         Diffuse $z_{t^{(k)}}^m$ using $\epsilon$; **Select Condition:** $c^m \leftarrow$ `SelfCond` $(k \leq 2)$ / `CrossCond` $(k=3)$.
10:         Denoise $(\hat{\epsilon}^m, mid^m) \leftarrow U_\theta(z_{t^{(k)}}^m, t^{(k)}, c^m)$ and recover $\hat{z}_0^m$ via Eq. 13.
11:     **end for**
12:     **Accumulate Objectives:**
13:         $\mathcal{L} += \{\lambda_2 \mathcal{L}_{\text{id}} + \lambda_3 \mathcal{L}_{\text{diff}} + \lambda_4 \mathcal{L}_{\text{sym}} + \lambda_5 \mathcal{L}_{\text{rec}}\}$.
14:     Update $\Theta$ via gradient descent on $\mathcal{L}$.
15: **until** converged

---

$\bar{b}^m \in \mathbb{R}^{(B \times W') \times C' \times H'}$. We then apply Multi-Head Self-Attention (MSA) over the dimension $H'$ and perform a residual update to obtain the bridging state:

$$mid_t^m = b^m + \text{Reshape}(\text{MSA}(\bar{b}^m)). \qquad (11)$$

This axial attention introduces a specific inductive bias that models the holistic columnar structure of the human body, avoiding the overfitting of local textures. This yields a stable, controllable intermediate representation $mid_t^m$ that complements the horizontal partitioning of the HPP.

### 3.3. Phased Dynamics Alignment

DiffCrossGait advances cross-modal alignment from static endpoint constraints to dynamic diffusion constraints. To realize this, we establish a stratified noise coordinate system within the latent diffusion process.

**Adaptive Noise Interval Partitioning.** Given that the signal retention rate $\bar{\alpha}_t$ decreases monotonically with $t$, we employ two hyperparameters, $\rho_a$ and $\rho_b$, to segment the temporal axis into three distinct phases. We define the boundary timesteps as:

$$ta = \min\{t : \bar{\alpha}_t \leq \rho_a\}, \quad t_b = \min\{t : \bar{\alpha}_t \leq \rho_b\}, \qquad (12)$$

subject to $1 \leq t_a \leq t_b < T$. This stratification yields three non-overlapping intervals corresponding to Small $[1, t_a]$, Medium $(t_a, t_b]$, and Large $(t_b, T]$ noise levels. Unlike fixed step indices, this partitioning is derived entirely from the schedule function, facilitating robust adaptation to varying total steps $T$ or noise schedules and eliminating the arbitrariness of manual heuristics.

**Stratified Sampling and Denoising.** In each training iteration, we independently sample timesteps from the three defined regimes: $t^{(1)} \in [1, t_a]$, $t^{(2)} \in (t_a, t_b]$, and $t^{(3)} \in (t_b, T]$. For each sampled timestep, we apply the forward diffusion process (Eq. 7) to both modalities, driven by the shared noise source. The shared denoising network subsequently predicts the noise component and extracts the bottleneck bridging state (Eq. 8) conditioned on the corresponding high-level semantics. Furthermore, we derive an estimate of the clean latent state directly from the noise prediction via:

$$\hat{z}_{0,t}^m = \frac{1}{\sqrt{\bar{\alpha}_t}}(z_t^m - \sqrt{1 - \bar{\alpha}_t}\hat{\epsilon}_t^m). \qquad (13)$$

This operation yields the reconstructed latents for each phase: $\hat{z}_{0,t^{(1)}}^m$, $\hat{z}_{0,t^{(2)}}^m$, and $\hat{z}_{0,t^{(3)}}^m$. We then impose distinct alignment constraints for each phase as follows: **Phase1: Identity Anchoring (Small Noise).** At $t^{(1)}$, the noise perturbation is minimal, and the underlying identity structure remains largely intact. We employ `SelfCond` routing and impose explicit identity supervision on the estimated clean latent $\hat{z}_{0,t^{(1)}}^m$ to prevent semantic drift. This constraint ensures that the diffusion branch learns a faithful restoration trajectory, preserving discriminative fidelity rather than inducing manifold distortion. Analogous to the discriminative branch, we project $\hat{z}_{0,t^{(1)}}^m$ through the shared prediction heads (SharedFCs and SharedBNNecks) to extract embeddings $e_{id}^m$ and logits $l_{id}^m$. The identity anchoring loss $\mathcal{L}_{\text{id}}$ is then computed using the formulation defined in Eq. 5. **Phase2: Dynamics Consistency (Medium Noise).** In the medium noise regime $t^{(2)}$, applying direct identity supervision risks inducing overfitting to superficial discriminative cues rather than learning robust latent dynamics. Consequently, we shift our focus to constraining the denoising mechanism itself. We enforce consistency in the noise prediction to ensure both modalities yield a unified interpretation of the shared perturbation, and we align the bottleneck bridging states to synchronize the trajectory's key intermediate representations:

$$\mathcal{L}_{\text{diff}}^{t^{(2)}} = \frac{1}{2}\left(|\hat{\epsilon}_{t^{(2)}}^{2d} - \epsilon|_2^2 + |\hat{\epsilon}_{t^{(2)}}^{3d} - \epsilon|_2^2\right), \qquad (14)$$

$$\mathcal{L}_{\text{sym}}^{t^{(2)}} = |mid_{t^{(2)}}^{2d} - mid_{t^{(2)}}^{3d}|_2^2. \qquad (15)$$

To mitigate potential trivial solutions (e.g., representation collapse) arising from pure alignment constraints, we incorporate an intra-modal $L_1$ reconstruction loss, which regresses the estimated clean latent back to its modality-specific ground truth:

$$\mathcal{L}_{\text{rec}-\text{self}} = |\hat{z}_{0,t^{(2)}}^{2d} - z_0^{2d}|_1 + |\hat{z}_{0,t^{(2)}}^{3d} - z_0^{3d}|_1. \qquad (16)$$

This term imposes an explicit reversibility constraint, balancing dynamic consistency with structural reconstruction fidelity. **Phase3: Cross-Modal Alignment (Large Noise).** In the large noise regime $t^{(3)}$, the latent state $z_t^m$ approximates an isotropic Gaussian distribution, and the denoising process relies heavily on the structural priors provided by the condition vector. Consequently, we transition to `CrossCond` routing. This mechanism compels the two modalities to establish a reciprocal structural recovery mechanism under the shared parameters of $U_\theta$. Accordingly, we compute $\mathcal{L}_{\text{diff}}^{t^{(3)}}$ and $\mathcal{L}_{\text{sym}}^{t^{(3)}}$ following the formulation in Phase2. Furthermore, we introduce an inter-modal $L_1$ reconstruction loss, which targets the clean latent of the counterpart modality:

$$\mathcal{L}_{\text{rec-cross}} = |\hat{z}_{0,t^{(3)}}^{2d} - z_0^{3d}|_1 + |\hat{z}_{0,t^{(3)}}^{3d} - z_0^{2d}|_1. \quad (17)$$

In contrast to the self-reconstruction of Phase2, $\mathcal{L}_{\text{rec-cross}}$ enforces alignment at the prior level under strong noise perturbation: the denoised output must not only recover valid structural properties but also converge to a consistent manifold defined by the cross-modal anchor, thereby significantly narrowing the domain gap.

### 3.4. Overall Optimization and Efficient Inference

In summary, we optimize the framework by concurrently supervising the discriminative and generative branches for cross-modal recognition. The total objective function for DiffCrossGait is summarized in Alg. 1 and formulated as:

$$\begin{aligned} \mathcal{L} = &\lambda_1 \mathcal{L}_{\text{disc}} + \lambda_2 \mathcal{L}_{\text{id}} \\ &+ \lambda_3 \left( \mathcal{L}_{\text{diff}}^{t^{(2)}} + \mathcal{L}_{\text{diff}}^{t^{(3)}} \right) + \lambda_4 \left( \mathcal{L}_{\text{sym}}^{t^{(2)}} + \mathcal{L}_{\text{sym}}^{t^{(3)}} \right) \\ &+ \lambda_5 \left( \mathcal{L}_{\text{rec-self}} + \mathcal{L}_{\text{rec-cross}} \right), \end{aligned} \quad (18)$$

where the hyperparameters are empirically set to $\lambda_1 = 1.0, \lambda_2 = 0.5, \lambda_3 = \lambda_4 = 0.1, \lambda_5 = 0.05$. In the inference time, we exclusively utilize the feature embeddings from the discriminative backbone for recognition.

## 4. Experiments

### 4.1. Datasets and Evaluation Protocols

We evaluate DiffCrossGait on two cross-modal LiDAR–camera gait benchmarks: SUSTech1K (Shen et al., 2023) and FreeGait (Han et al., 2024). Following standard protocols, we report Rank-1 and Rank-5 accuracy (%) for two retrieval directions: **2D→3D** and **3D→2D**. In addition to overall accuracy, we provide a breakdown across eight covariate conditions to stress-test robustness under various conditions, including appearance shifts, partial occlusion, and illumination changes. Details of the datasets and the implementations are provided in the Appendix A.1 and A.2.

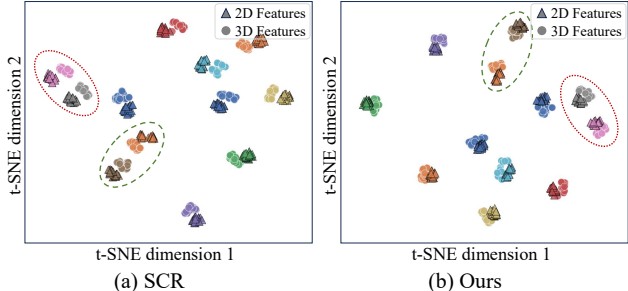

(a) SCR  (b) Ours

*Figure 4.* **Comparison of t-SNE Visualizations.** Compared to SCR, our proposed method yields more compact intra-class clusters and greater inter-class distances across different modalities. The dashed boxes highlight the changes in the intra-class and inter-class clusters for the corresponding IDs.

### 4.2. Comparative Analysis

**Baseline.** We compare against representative cross-modal gait methods (e.g., CrossGait (Wang et al., 2024) and CL-Gait (Guo et al., 2025)), early LiDAR/cross-modal pipelines (e.g., CAJ (Ye et al., 2021), SAAI (Fang et al., 2023), Lidar-Gait (Shen et al., 2023)), and competitive visible-infrared ReID models adapted to cross-modal retrieval (IDKL (Ren & Zhang, 2024), TVI-LFM (Hu et al., 2024), TSKD (Shi et al., 2026), SCR (Yu et al., 2025a)).

**Main results on SUSTech1K (2D→3D).** Tab. 1 shows that DiffCrossGait establishes a new state-of-the-art in the Camera→LiDAR retrieval setting, which outperforms the strongest reported competitors by +3.6 Rank-1 and +1.4 Rank-5. Beyond the overall gain, DiffCrossGait improves consistently across the majority of covariates. These cases are precisely where endpoint-based discriminative alignment is fragile. DiffCrossGait mitigates this by aligning the denoising dynamics rather than only the final embedding. Empirically, this phased constraint acts like a curriculum over noise scales, encouraging invariances that survive modality changes and covariate shifts. The only exception is **Occlusion**, where DiffCrossGait is slightly lower than SCR (68.4 vs. 69.4, i.e., **-1.0** Rank-1), indicating that extreme missing-body evidence remains challenging even under trajectory regularization; nevertheless, the method remains competitive (second-best) while maintaining stronger overall robustness.

**Main results on SUSTech1K (3D→2D).** In the reverse retrieval direction (LiDAR→Camera), Tab. 2 shows an even larger advantage. DiffCrossGait exceeds the best baseline by +6.1 Rank-1 and +3.6 Rank-5, with substantial gains on conditions that induce large intra-class variance: Normal (+11.3), Bag (+11.4), Uniform (+11.5), Carrying (+9.3), Umbrella (+8.2), and Clothing (+8.5) in Rank-1. On the SUSTech1K test set, we qualitatively and quantitatively highlight the cross-modal recognition capability of DiffCrossGait via visualization (Van der Maaten & Hinton,

*Table 1.* **Quantitative Comparison on SUSTech1K (2D Camera → 3D LiDAR).** We report Rank-1 and Rank-5 accuracy (%). ♠ denotes methods requiring extra synthetic data for pre-training. Our DiffCrossGait achieves SOTA performance in most covariates, particularly in challenging scenarios like *Night* and *Bag*.

| Method | | Covariate Conditions (Rank-1) | | | | | | | | Overall(%) | |
|---|---|---|---|---|---|---|---|---|---|---|---|
| | | Normal | Bag | Clothing | Carrying | Umbrella | Uniform | Occlusion | Night | Rank-1 | Rank-5 |
| CAJ (Ye et al., 2021) | ICCV'21 | 16.4 | - | 7.5 | - | 7.4 | - | - | 2.4 | 11.3 | 30.1 |
| SAAI (Fang et al., 2023) | ICCV'23 | 22.4 | - | 14.3 | - | 14.0 | - | - | 5.3 | 23.1 | 49.5 |
| LidarGait (Shen et al., 2023) | CVPR'23 | 18.2 | - | 3.4 | - | 3.4 | - | - | 4.7 | 9.6 | 28.1 |
| CL-Gait♠ (Guo et al., 2025) | ECCV'24 | - | - | - | - | - | - | - | - | 55.1 | 77.3 |
| CrossGait (Wang et al., 2024) | IJCB'24 | 63.2 | - | 30.6 | - | 38.5 | - | - | 11.8 | 53.6 | 77.0 |
| IDKL (Ren & Zhang, 2024) | CVPR'24 | 60.3 | 49.8 | 29.2 | 48.5 | 36.9 | 50.7 | 64.2 | 9.4 | 52.2 | 75.2 |
| TVI-LFM (Hu et al., 2024) | NeurIPS'24 | 61.0 | 50.3 | 30.1 | 50.2 | 37.5 | 51.0 | 66.5 | 10.0 | 53.0 | 76.1 |
| TSKD (Shi et al., 2026) | PR'25 | 52.1 | 43.6 | 27.9 | 48.0 | 32.7 | 41.1 | 55.6 | 6.3 | 42.6 | 65.8 |
| SCR (Yu et al., 2025a) | IF'25 | 61.3 | 52.9 | 29.6 | 53.0 | 39.1 | 53.7 | **69.4** | 10.3 | 54.9 | 78.1 |
| **DiffCrossGait (Ours)** | - | **70.6**↑7.4 | **62.3**↑9.4 | **37.9**↑7.3 | **58.5**↑5.5 | **44.1**↑5.0 | **56.7**↑3.0 | 68.4↓1.0 | **12.7**↑0.9 | **58.7**↑3.6 | **79.5**↑1.4 |

*Table 2.* **Quantitative Comparison on SUSTech1K (3D LiDAR → 2D Camera).** ♠ denotes methods pre-trained on synthetic data. Our method outperforms SOTA competitors by a large margin in most conditions, establishing a new benchmark for cross-modal retrieval.

| Method | | Covariate Conditions (Rank-1) | | | | | | | | Overall(%) | |
|---|---|---|---|---|---|---|---|---|---|---|---|
| | | Normal | Bag | Clothing | Carrying | Umbrella | Uniform | Occlusion | Night | Rank-1 | Rank-5 |
| CAJ (Ye et al., 2021) | ICCV'21 | 15.3 | - | 6.4 | - | 13.0 | - | - | 2.3 | 12.3 | 32.3 |
| SAAI (Fang et al., 2023) | ICCV'23 | 26.5 | - | 21.9 | - | 23.2 | - | - | 3.2 | 26.1 | 54.1 |
| LidarGait (Shen et al., 2023) | CVPR'23 | 23.2 | - | 14.2 | - | 24.7 | - | - | 2.4 | 18.3 | 39.6 |
| CL-Gait♠ (Guo et al., 2025) | ECCV'24 | - | - | - | - | - | - | - | - | 53.3 | 75.6 |
| CrossGait (Wang et al., 2024) | IJCB'24 | 62.2 | - | 35.4 | - | 57.8 | - | - | **10.3** | 56.4 | 79.8 |
| IDKL (Ren & Zhang, 2024) | CVPR'24 | 59.6 | 52.3 | 31.0 | 49.5 | 55.2 | 56.1 | 65.3 | 7.9 | 54.8 | 77.1 |
| TVI-LFM (Hu et al., 2024) | NeurIPS'24 | 60.4 | 53.0 | 32.7 | 51.6 | 56.4 | 55.8 | 69.2 | 9.1 | 55.7 | 78.5 |
| TSKD (Shi et al., 2026) | PR'25 | 50.1 | 41.3 | 27.7 | 42.8 | 45.9 | 46.2 | 52.5 | 7.8 | 47.2 | 68.1 |
| SCR (Yu et al., 2025a) | IF'25 | 61.6 | 54.1 | 35.8 | 52.0 | 58.1 | 55.9 | 72.6 | 10.2 | 57.7 | 79.5 |
| **DiffCrossGait (Ours)** | - | **73.5**↑11.3 | **65.5**↑11.4 | **44.3**↑8.5 | **61.3**↑9.3 | **66.3**↑8.2 | **67.6**↑11.5 | **73.5**↑0.9 | 8.1↓2.2 | **63.8**↑6.1 | **83.4**↑3.6 |

*Table 3.* **Quantitative Comparison on FreeGait Dataset.** Our DiffCrossGait achieves significant performance gains (+10%∼18%) on this large-scale benchmark, demonstrating robust generalization.

| Method | | 2D → 3D(%) | | 3D → 2D(%) | |
|---|---|---|---|---|---|
| | | Rank-1 | Rank-5 | Rank-1 | Rank-5 |
| HMRNet (Han et al., 2024) | MM'24 | 23.5 | 55.7 | 25.1 | 57.0 |
| CrossGait (Wang et al., 2024) | IJCB'24 | 29.6 | 60.8 | 32.3 | 65.9 |
| IDKL (Ren & Zhang, 2024) | CVPR'24 | 36.7 | 67.4 | 39.5 | 70.3 |
| TVI-LFM (Hu et al., 2024) | NeurIPS'24 | 38.9 | 69.1 | 41.0 | 71.8 |
| TSKD (Shi et al., 2026) | PR'25 | 25.1 | 57.9 | 26.7 | 60.8 |
| SCR (Yu et al., 2025a) | IF'25 | 40.1 | 72.0 | 43.3 | 75.9 |
| **DiffCrossGait (Ours)** | - | **58.5**↑18.4 | **86.9**↑14.9 | **61.5**↑18.2 | **86.8**↑10.9 |

*Table 4.* **Inference Efficiency Analysis.** Comparison of model complexity and inference speed. Note that DiffCrossGait discards the diffusion branch during inference, maintaining the same computational cost as the baseline backbone while significantly boosting performance.

| Method | Inference Complexity* | | | Performance |
|---|---|---|---|---|
| | Params (M) | FLOPs (G) | Time (ms) | Rank-1 (%) |
| IDKL (Ren & Zhang, 2024) | 74.1 | 1.1 | 10.5 | 54.8 |
| SCR (Yu et al., 2025a) | 117.8 | 0.9 | 11.7 | 57.7 |
| Baseline (ResNet-9) | **13.7** | 1.5 | **5.9** | 54.7 |
| **DiffCrossGait (Ours)** | **14.0** | 1.4 | 6.2 | **63.8 (+9.1)** |

\* Testing on a single NVIDIA RTX 3090 GPU.

2008), as shown in Fig. 4 and Fig. 5, respectively.

**Generalization to real-world scenarios (FreeGait).** Tab. 3 demonstrates that DiffCrossGait generalizes strongly to FreeGait, which is collected in unconstrained environments with complex illumination and unstructured occlusions. DiffCrossGait achieves more superior results compared to the strongest baseline SCR (Yu et al., 2025a). Such gains are difficult to achieve via static alignment alone, supporting the hypothesis that diffusion-trajectory constraints encourage modality-invariant structures that are less tied to

dataset-specific sensor artifacts, leading to improved out-of-distribution robustness.

### 4.3. Efficiency and Practicality Considerations

A key design choice in DiffCrossGait is that diffusion is used only as a training-time objective; inference relies solely on the discriminative backbone. As shown in Tab. 4, DiffCrossGait retains essentially the same inference footprint as the baseline backbone, while delivering a large accuracy gain. In contrast, stronger discriminative baselines

*Table 5.* **Ablation of Key Mechanisms and Architectural Components.** We verify the necessity of the unified trajectory design and the lightweight architecture. *Shared Noise* denotes driving both modalities with identical Gaussian noise.

| Trajectory Mechanism | | Architecture | | Metric (%) | |
|---|---|---|---|---|---|
| Shared Noise | Decoupled Heads | Shared Modal Token | Axial Attention | Rank-1 | Rank-5 |
| | ✓ | ✓ | ✓ | 61.8 | 81.9 |
| ✓ | | ✓ | ✓ | 62.7 | 82.7 |
| ✓ | ✓ | | ✓ | 60.3 | 81.0 |
| ✓ | ✓ | ✓ | | 62.0 | 82.1 |
| ✓ | ✓ | ✓ | ✓ | **63.8** | **83.4** |

*Table 6.* **Impact of the Tri-Phase Alignment Strategy**. We utilize varying noise intensities to impose hierarchical constraints. Phase1: Small noise interval (Identity Anchoring); Phase2: Medium noise interval (Dynamics Consistency); Phase3: Large noise interval (Cross-Modal Alignment).

| Exp. | Noise Intervals & Constraints | | | Metric (%) | |
|---|---|---|---|---|---|
| | Phase1 | Phase2 | Phase3 | Rank-1 | Rank-5 |
| Base | | | | 54.7 | 76.5 |
| 1 | ✓ | | | 60.6 | 81.0 |
| 2 | ✓ | ✓ | | 63.1 | 82.9 |
| 3 | ✓ | | ✓ | 62.5 | 82.6 |
| 4 | | ✓ | ✓ | 61.8 | 82.0 |
| Ours | ✓ | ✓ | ✓ | **63.8** | **83.4** |

such as IDKL and SCR incur higher parameter counts and slower inference. This suggests that the performance gains of DiffCrossGait are not achieved by scaling inference-time computation, but rather by enhancing training-time cross-modal alignment through a lightweight auxiliary diffusion branch and trajectory-level constraints, which is attractive for deployment-oriented gait recognition systems. During training, the auxiliary diffusion branch adds 2.3M parameters and increases the training time from 8.5h to 11.2h on a single RTX 3090; however, this branch is discarded after training and introduces negligible inference overhead.

### 4.4. Ablation Study

**Necessity of the unified trajectory design and lightweight components.** Tab. 5 isolates four key elements in DiffCrossGait. Removing **Shared Noise** drops performance, verifying that a synchronized stochastic driver is essential for process alignment. Without shared $\epsilon$, the two modalities experience unrelated perturbations, and the denoising objective can be satisfied independently per modality, weakening the cross-modal constraint on the trajectory geometry. We further observe that the **Shared Modal Token** is the most critical architectural factor: removing it reduces accuracy. In our framework, the shared modal token also operationalizes the conditioning design used by SelfCond/CrossCond routing: it enables the same denoiser to switch between self-driven restoration and cross-driven recoverability, which is diffi-

*Table 7.* **Ablation of the Denoising U-Net Architecture.** We investigate the impact of weight sharing and model capacity on recognition performance and training cost. *Indep.*: Independent U-Nets for 2D/3D; *Heavy*: Standard deep U-Net structure.

| U-Net Configuration | | Training Cost | | Metric (%) | |
|---|---|---|---|---|---|
| **Weight Sharing** | **Model Size** | **Params (M)** | **Time (h)\*** | **Rank-1** | **Rank-5** |
| Independent | Lightweight | ~1.2× | ~1.1× | 63.3 | 83.1 |
| Shared | Heavyweight | ~2.6× | ~1.7× | 59.1 | 79.8 |
| **Shared (Ours)** | **Lightweight** | **1×** | **1×** | **63.8** | **83.4** |

\* Training on a single NVIDIA RTX 3090 GPU.

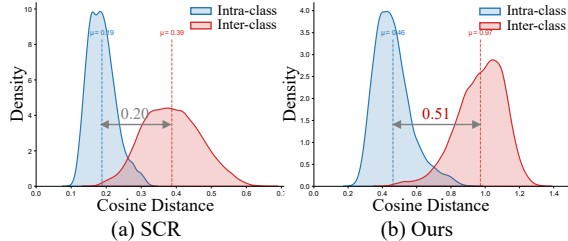

*Figure 5.* **Visualization of intra-class and inter-class cosine distances.** Compared with SCR, our method exhibits a larger inter–intra distance gap, indicating stronger separability.

cult to emulate with identity priors alone. The **Decoupled Heads** and **Axial Attention** each provide consistent gains. When disabling head decoupling, performance decreases, supporting the motivation that separating discriminative and generative projections reduces optimization interference. Removing axial attention degrades the accuracy, consistent with the interpretation that axial attention injects an efficient global dependency bias into the bottleneck state, improving the controllability of intermediate representations that are explicitly aligned across modalities.

**Impact of the Tri-Phase Alignment Strategy.** Tab. 6 validates the proposed curriculum over noise regimes. Adding only **Phase1** improves the baseline performance, showing that anchoring near clean latents effectively prevents semantic drift in the diffusion branch and preserves discriminative integrity. Introducing **Phase2** on top of Phase1 further increases performance, suggesting that aligning denoising dynamics (noise prediction and bottleneck bridging states) provides a stronger cross-modal coupling signal than identity supervision alone once the signal-to-noise ratio decreases. **Phase3** complements the above by enforcing cross-modal structural recoverability under large noise. While Phase1+3 (62.5/82.6) and Phase2+3 (61.8/82.0) already outperform the baseline substantially, the full strategy achieves the best result (63.8/83.4), improving over Phase1+2 by an additional +0.7 Rank-1. This pattern supports the intended semantics of the three regimes: Phase1 maintains class-discriminative structure locally; Phase2 synchronizes the trajectory's evolution in the mid-noise region, where direct identity losses can be overly restrictive; and Phase3 imposes

*Table 8.* Sensitivity analysis on SUSTech1K. Loss-weight stability is evaluated under the stronger reverse retrieval setting (3D→2D), while schedule robustness is evaluated under 2D→3D by varying phase boundaries and diffusion length.

| Diagnostic | Setting | Rank-1 |
|---|---|---|
| Loss weights $(\lambda_2, \lambda_3, \lambda_5)$ | $(0.1, 0.1, 0.05)$ | 62.1 |
| | $(1.0, 0.1, 0.05)$ | 63.3 |
| | $(0.5, 0.05, 0.05)$ | 62.9 |
| | $(0.5, 0.2, 0.1)$ | 63.1 |
| | $(0.5, 0.1, 0.05)$ | **63.8** |
| $\rho_a$ with $\rho_b = 0.50$ | 0.90 / 0.95 / 0.98 | 58.3 / **58.7** / 58.5 |
| $\rho_b$ with $\rho_a = 0.95$ | 0.40 / 0.50 / 0.60 | 58.1 / **58.7** / 58.4 |
| Diffusion length $T$ | 50 / 100 / 200 | 58.2 / **58.7** / 58.6 |

a global prior that narrows the modality gap when both latents are close to a Gaussian and recovery must rely on structured conditions rather than modality-specific cues.

**Ablation of the denoising U-Net design.** Tab. 7 shows that our denoiser should be shared and lightweight. Using independent lightweight U-Nets slightly below the shared lightweight design while increasing training cost to ∼1.2× parameters and ∼1.1× time. This indicates that parameter sharing itself acts as an implicit alignment regularizer: it constrains both modalities to admit compatible denoising transformations under the same function class, reinforcing the notion of a unified trajectory. Conversely, a shared heavyweight U-Net substantially degrades performance despite requiring ∼2.6× parameters and ∼1.7× training time. A plausible explanation is that excessive denoising capacity allows the diffusion branch to "explain away" modality-specific artifacts independently, thereby weakening the pressure to learn modality-invariant structure in the bottleneck states and along the trajectory.

### 4.5. Sensitivity, Generalizability, and Diagnostics

**Sensitivity.** We evaluate the stability of DiffCrossGait under different loss weights, phase boundaries, and diffusion lengths. As shown in Tab. 8, the performance changes mildly around our default setting, indicating that the gain is not caused by a fragile hyperparameter choice. The same loss weights are used on both SUSTech1K (Shen et al., 2023) and FreeGait (Han et al., 2024).

**Generalizability.** To examine whether the diffusion objective is tied to a specific gait backbone, we further apply it to two visible-infrared ReID baselines on SYSU-MM01 (Wang et al., 2019) under the all-search single-shot protocol. DEEN (Zhang & Wang, 2023) improves from 74.70/71.80 to 75.19/72.01 in Rank-1/mAP, and IDKL (Ren & Zhang, 2024) improves from 81.42/79.85 to 81.91/80.11. Although modest, these consistent gains suggest that trajectory-level latent regularization can be used as a plug-and-play objective for heterogeneous representation learning.

**Trajectory diagnostic.** We also measure the gap between predicted clean states, $\|\hat{z}_{0,t}^{2d} - \hat{z}_{0,t}^{3d}\|_2$, which better reflects learned alignment than directly comparing noisy latents. DiffCrossGait reduces the endpoint gap from 0.23 to 0.12 and keeps the trajectory gap controlled at $t=10, 50, 90$ as 0.18, 0.31, and 0.63, whereas a Phase1-only variant increases from 0.34 at $t=10$ to 1.74 at $t=90$. This indicates that the multi-phase design actively suppresses cross-modal divergence under severe noise.

### 4.6. Failure Cases and Limitations

DiffCrossGait can still fail under extreme information scarcity, e.g., when nighttime degradation and heavy occlusion or carrying occur simultaneously. In such cases, both modalities may lose essential identity-related structural anchors, making the shared clean latent state difficult to recover. Another limitation comes from the axial attention prior: it assumes an approximately vertical head-to-toe body layout, which is suitable for ground-level horizontal viewpoints but may break under extreme pitch angles such as top-down UAV scenarios. Qualitative retrieval comparisons with SCR and representative failure cases are provided in Appendix A.5 and Appendix A.6, respectively.

## 5. Conclusion

We propose DiffCrossGait, a diffusion-based trajectory alignment framework for 2D–3D cross-modal gait recognition that couples Camera and LiDAR features with shared Gaussian noise in a unified latent diffusion process. A Tri-Phase Alignment Strategy enforces complementary constraints across noise regimes—identity anchoring, denoising/bottleneck consistency, and cross-modal recoverability via cross-conditioning—yielding modality-invariant representations. Experiments demonstrate consistent state-of-the-art performance, and these results suggest that shared-noise diffusion is most effective as a training-time identity-subspace regularizer when both modalities retain sufficient gait-related structural evidence.

## Impact Statement

This work advances cross-modal gait recognition, a remote biometric technology that may support identification when face or fingerprint signals are unavailable. However, such systems also raise risks of mass surveillance, non-consensual tracking, privacy infringement, and biased deployment across environments or populations. DiffCrossGait is a representation-learning method and does not by itself mitigate these risks. Any deployment should require legal authorization, appropriate consent, strict access control, auditable data governance, and evaluation under the target distribution. Although our diffusion objective focuses on

structural gait dynamics rather than appearance textures, it should not be interpreted as a privacy-preserving guarantee.

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

# A. Appendix

### A.1. Datasets and Evaluation Protocols

We evaluate our proposed framework on two widely used cross-modal LiDAR-Camera datasets: SUSTech1K (Shen et al., 2023) and FreeGait (Han et al., 2024). SUSTech1K comprises 25,239 sequences from 1,050 subjects recorded from 12 distinct views. To simulate diverse gait recognition scenarios, various covariates are incorporated, including clothing changes, occlusions, carrying conditions, and illumination variations. Following standard protocols (Wang et al., 2024; Guo et al., 2025; Shen et al., 2025; 2023), the dataset is partitioned into a training set of 250 identities (6,011 sequences) and a test set of 800 identities (19,228 sequences), providing a comprehensive benchmark for cross-modal validation. FreeGait is a large-scale dataset collected in real-world wild scenarios within a range of 25 meters. It contains 11,921 sequences from 1,195 identities, capturing complex environmental factors such as dynamic illumination and unstructured occlusions. The dataset is split into 500 identities for training and 695 for testing, serving as a rigorous testbed for model robustness in practical applications (Han et al., 2024; Shen et al., 2025). For evaluation, we report the Rank-1 and Rank-5 accuracy metrics separately for both retrieval directions: Camera-to-LiDAR (2D→3D) and LiDAR-to-Camera (3D→2D). We also provide a detailed performance breakdown across different covariate conditions to assess the efficacy of our method under varying challenges.

### A.2. Implementation Details

Following prior methodologies (Guo et al., 2025; Wang et al., 2024), we utilize silhouettes and range views as the input representations for the 2D Camera and 3D LiDAR modalities, respectively. For the diffusion process, the total number of timesteps is set to $T = 100$, with a timestep embedding dimension of 256. We employ a weight-shared U-Net as the denoising network for both modality branches to facilitate implicit alignment. The denoising network is a lightweight hybrid U-Net with four ResBlocks and one axial-attention bottleneck, operating over $16 \times 16$ to $4 \times 4$ feature scales with channel dimensions 512 and 256. The adaptive thresholds for our Tri-Phase Alignment Strategy are empirically set to $\rho_a = 0.95$ and $\rho_b = 0.5$. We utilize a TripletSampler to construct mini-batches. Each batch consists of 8 randomly sampled identities, with 8 sequences per identity, resulting in a total batch size of 64. For each sequence, we strictly sample a fixed length of 10 frames. The model is optimized using the Adam optimizer with an initial learning rate of $7 \times 10^{-4}$ and a weight decay of $5 \times 10^{-4}$. We employ a MultiStep learning rate scheduler, decaying the learning rate by a factor of 0.1 at 15,000 and 30,000 iterations, with a total training duration of 40,000 iterations. All experiments are implemented in PyTorch and conducted on a single NVIDIA RTX 3090 GPU.

### A.3. Input Representation Discussion

We follow the standard silhouette/range-view protocol because it balances structural compatibility and geometric fidelity. In a controlled experiment, converting LiDAR range views into pseudo-2D silhouettes reduces the baseline Rank-1 accuracy to 27.6% for 2D→3D and 28.2% for 3D→2D. Directly pairing raw sparse point clouds with 2D silhouettes further enlarges the modality gap, leading to 10.1% and 11.9% Rank-1, respectively. These results support using geometry-preserving range views and performing alignment in latent trajectory space.

### A.4. Theoretical Feasibility & Reliability Analysis

In this section, we provide a theoretical analysis of the proposed **DiffCrossGait** framework. We aim to rigorously demonstrate why the proposed **Shared-Noise Trajectory Alignment** provides a strictly stronger constraint for cross-modal manifold alignment compared to traditional **Static Endpoint Alignment**. We base our analysis on Stochastic Differential Equations (SDEs), Score Matching theory, and Domain Adaptation generalization bounds.

#### A.4.1. PRELIMINARIES AND NOTATION

Let $\mathcal{X}_{2d}$ and $\mathcal{X}_{3d}$ denote the input spaces for 2D silhouette sequences and 3D point cloud sequences, respectively. Let $f_\theta : \mathcal{X} \to \mathcal{Z}$ represent the backbone encoders, mapping inputs to a shared latent manifold $\mathcal{Z} \subseteq \mathbb{R}^d$. Let $p_{2d}(\mathbf{z})$ and $p_{3d}(\mathbf{z})$ denote the induced latent distributions for the two modalities.

The forward diffusion process is modeled as a discretized SDE. In the continuous limit, for a latent variable $\mathbf{z}(t)$, the forward SDE is given by:

$$d\mathbf{z} = \mathbf{f}(\mathbf{z}, t)dt + g(t)d\mathbf{w}, \quad t \in [0, T] \tag{19}$$

where $\mathbf{w}$ is a standard Wiener process (Brownian motion), $\mathbf{f}(\cdot)$ is the drift coefficient, and $g(t)$ is the diffusion coefficient. **DiffCrossGait** enforces that for paired samples $(\mathbf{x}_{2d}, \mathbf{x}_{3d})$, the noise realization $d\mathbf{w}$ (implemented as $\epsilon$ in discrete time) is *identical*.

### A.4.2. OPTIMIZATION STABILITY VIA LIPSCHITZ REGULARIZATION

A core challenge in cross-modal learning (e.g., Triplet Loss) is the instability of the optimization landscape, often leading to mode collapse where embedding manifolds intersect only at specific points rather than aligning globally. We argue that the auxiliary diffusion objective acts as a Lipschitz regularizer.

**Definition A.1** (Lipschitz Continuity of the Encoder). An encoder $f_\theta$ is $K$-Lipschitz if $\|f_\theta(x) - f_\theta(y)\| \leq K \|x - y\|$ for all $x, y$.

**Theorem A.2** (Smoothness of the Optimization Landscape). *Let $\mathcal{L}_{diff}(\mathbf{z})$ be the denoising score matching loss defined on the latent code $\mathbf{z}$. Assuming the denoising network $U_\theta$ approximates the score function $\nabla_{\mathbf{z}} \log p_t(\mathbf{z})$ with bounded error, minimizing $\mathcal{L}_{diff}$ constrains the trace of the Hessian of the log-likelihood of the latent distribution, thereby bounding the local curvature of the optimization landscape.*

*Proof Sketch.* The diffusion loss in DiffCrossGait can be viewed as learning the score function $s_\theta(\mathbf{z}, t) \approx \nabla_{\mathbf{z}} \log p_t(\mathbf{z})$. Vincent showed that Denoising Autoencoder objectives are equivalent to regularizing the Frobenius norm of the Jacobian of the mapping (Vincent, 2011). Specifically, for a Gaussian kernel with variance $\sigma^2$:

$$\mathcal{L}_{\text{diff}} \approx \mathbb{E}_{\mathbf{z}} \left[ \|\mathbf{z} - r(\mathbf{z})\|^2 \right] \propto \text{Tr}\left( \nabla^2 \log p(\mathbf{z}) \right) + o(\sigma^2) \tag{20}$$

where $r(\mathbf{z})$ is the reconstruction. By minimizing $\mathcal{L}_{\text{diff}}$ on the latent codes $z_{2d}$ and $z_{3d}$, we explicitly penalize high-frequency variations in the latent density $p(\mathbf{z})$. This acts as a spectral normalizer on the Jacobian $J_{f_\theta} = \frac{\partial \mathbf{z}}{\partial \mathbf{x}}$, ensuring that:

$$\sup_{\mathbf{x}} \|J_{f_\theta}(\mathbf{x})\|_2 \leq C \tag{21}$$

This constraint prevents the encoder from learning "shortcuts" that map disparate 2D/3D inputs to the same point via highly distorted, non-smooth mappings, thus ensuring a smoother convergence basin for the discriminative backbone. $\qquad\square$

### A.4.3. TRAJECTORY ALIGNMENT VS. ENDPOINT ALIGNMENT

This is the central theoretical contribution. We show that alignment via shared noise trajectories bounds the **Fisher Divergence**, which is a stronger condition than minimizing Euclidean distance between endpoints.

**Lemma A.3** (Equivalence to Score Matching). *Minimizing the noise prediction error $\|\epsilon - \hat{\epsilon}_\theta(\mathbf{z}_t, t)\|^2$ is equivalent to minimizing the Fisher Divergence between the data distribution and the model distribution (Song et al., 2021b):*

$$\mathcal{L}_{SM} = \frac{1}{2} \int_0^T g(t)^2 \mathbb{E}_{p_t(\mathbf{z})} \left[ \|\nabla_{\mathbf{z}} \log p_t(\mathbf{z}) - s_\theta(\mathbf{z}, t)\|^2 \right] dt \tag{22}$$

**Theorem A.4** (Trajectory Consistency Implies Distributional Alignment). *Let $\mathbf{z}_{2d}$ and $\mathbf{z}_{3d}$ be latent representations of the same identity. Traditional endpoint alignment minimizes $\mathcal{L}_{end} = \|\mathbf{z}_{2d} - \mathbf{z}_{3d}\|^2$. The DiffCrossGait trajectory alignment minimizes the gap between their induced score fields. If the shared denoising trajectory loss approaches zero, then the Kullback-Leibler (KL) divergence between the conditional distributions $p(\mathbf{z}_t|\mathbf{x}_{2d})$ and $p(\mathbf{z}_t|\mathbf{x}_{3d})$ is bounded by the integrated trajectory error.*

*Proof Sketch.* Consider the shared diffusion process. The evolution of the probability density flow is governed by the Fokker-Planck equation. The KL divergence between the distributions of the two modalities at time $t = 0$ can be bounded using the path integrals of their score functions.

Let $\mathbf{s}_{2d}(\mathbf{z}, t) = \nabla \log p_t(\mathbf{z}|\mathbf{x}_{2d})$ and $\mathbf{s}_{3d}(\mathbf{z}, t) = \nabla \log p_t(\mathbf{z}|\mathbf{x}_{3d})$. The DiffCrossGait objective $\mathcal{L}_{diff}^{shared}$ enforces a shared denoiser $U_\theta$. Effectively, it minimizes the expected distance between the true scores of both modalities and the shared approximation:

$$\min_\theta \mathbb{E}_t \left[ \|\mathbf{s}_{2d}(\mathbf{z}, t) - U_\theta(\mathbf{z}, t)\|^2 + \|\mathbf{s}_{3d}(\mathbf{z}, t) - U_\theta(\mathbf{z}, t)\|^2 \right] \tag{23}$$

By the triangle inequality, this implies minimizing the **Fisher Divergence** between the two modalities:

$$\mathcal{J}_{Fisher}(p_{2d}\|p_{3d}) = \int \|\nabla \log p_{2d}(\mathbf{z}) - \nabla \log p_{3d}(\mathbf{z})\|^2 p(\mathbf{z})d\mathbf{z} \tag{24}$$

The Log-Sobolev inequality relates Fisher Divergence to KL Divergence:

$$KL(p_{2d}\|p_{3d}) \leq \frac{C}{2}\mathcal{J}_{Fisher}(p_{2d}\|p_{3d}) \tag{25}$$

**Crucial Distinction:** Endpoint alignment ($\mathcal{L}_{\text{end}}$) only constrains the zeroth-order moment (the means) of the distributions. Trajectory alignment ($\mathcal{L}_{\text{diff}}$), by matching the gradients of the log-density (the score) via shared noise $\epsilon$ across all $t$, constrains the **entire geometry of the distribution**. Thus, minimizing the trajectory error guarantees alignment of the underlying manifolds, whereas endpoint alignment can be satisfied by trivial solutions (e.g., mapping everything to a single point) that do not preserve geometric structure. $\square$

*Remark A.5.* This explains why DiffCrossGait is robust to covariates. Covariates (like clothing) distort the manifold. By enforcing alignment of the *dynamics* (the vector field required to denoise), we force the backbone to extract invariant features that obey the same physical generation laws, regardless of the input modality.

### A.4.4. GENERALIZATION BOUND FOR CROSS-MODAL RETRIEVAL

Finally, we analyze the method through the lens of Domain Adaptation theory to prove generalization capability.

**Definition A.6** ($\mathcal{H}\Delta\mathcal{H}$-Divergence). (Ben-David et al., 2010) Let $\mathcal{D}_S$ (2D) and $\mathcal{D}_T$ (3D) be source and target distributions over $\mathcal{Z}$. The discrepancy is:

$$d_{\mathcal{H}\Delta\mathcal{H}}(\mathcal{D}_S, \mathcal{D}_T) = 2 \sup_{h \in \mathcal{H}} |P_{\mathbf{z}\sim\mathcal{D}_S}[h(\mathbf{z}) \neq 1] - P_{\mathbf{z}\sim\mathcal{D}_T}[h(\mathbf{z}) \neq 1]| \tag{26}$$

where $h$ is a domain discriminator.

**Theorem A.7** (Generalization Bound). *For a hypothesis space $\mathcal{H}$, the expected error on the target domain (3D) $\epsilon_T(h)$ is bounded by:*

$$\epsilon_T(h) \leq \epsilon_S(h) + \frac{1}{2}d_{\mathcal{H}\Delta\mathcal{H}}(\mathcal{D}_S, \mathcal{D}_T) + \lambda \tag{27}$$

*In DiffCrossGait, the shared-noise diffusion objective minimizes an upper bound of $d_{\mathcal{H}\Delta\mathcal{H}}(\mathcal{D}_S, \mathcal{D}_T)$.*

*Proof Sketch.* The term $d_{\mathcal{H}\Delta\mathcal{H}}$ measures the distinguishability of the two domains. In our framework, the diffusion branch acts as a conditional generative model $p_\theta(\mathbf{x}|\mathbf{z})$. The shared U-Net $U_\theta$ attempts to denoise both $\mathbf{z}_{2d}$ and $\mathbf{z}_{3d}$ using the *same* parameters and the *same* noise $\epsilon$.

If the distributions $\mathcal{D}_S$ and $\mathcal{D}_T$ were disjoint (misaligned), the optimal denoising directions for $\mathbf{z}_{2d}$ and $\mathbf{z}_{3d}$ would be orthogonal or contradictory for a given $\epsilon$. The loss $\mathcal{L}_{sym} + \mathcal{L}_{diff}$ forces:

$$\mathbb{E}[\|U_\theta(\mathbf{z}_{2d} + \epsilon) - U_\theta(\mathbf{z}_{3d} + \epsilon)\|^2] \to 0 \tag{28}$$

This implies that for any discriminator $h$ relying on structural features exposed by the diffusion process, the samples are indistinguishable. The shared diffusion process implicitly matches the **supports** of the two distributions. Therefore, optimizing the trajectory consistency explicitly reduces the domain divergence term $d_{\mathcal{H}\Delta\mathcal{H}}$, leading to a tighter bound on the target (3D) generalization error compared to methods that only minimize source risk $\epsilon_S$. $\square$

### A.5. Qualitative Retrieval Analysis

To complement the distribution-level visualizations in the main paper, we further conduct a retrieval-level comparison with SCR (Yu et al., 2025a) on a manually curated challenging subset. This subset contains hard cross-modal cases with compound covariates such as bags, umbrellas, clothing changes, and disrupted contours. As summarized in Tab. 9, DiffCrossGait obtains 29/40 Top-1 correct retrievals, while SCR obtains 17/40. This subset is used only for diagnostic analysis and is not intended to replace the standard benchmark evaluation reported in the main paper.

*Table 9.* Rank-1 retrieval summary on a manually curated challenging subset. The subset is used for qualitative diagnosis under compound covariate shifts.

| Method | Rank-1 Correct | Total Cases |
|---|---|---|
| SCR | 17 | 40 |
| DiffCrossGait | **29** | 40 |

Fig. 6 shows two representative retrieval examples, covering both 2D→3D and 3D→2D directions. These examples illustrate a common pattern observed in the challenging subset: SCR is more likely to confuse identities when contours are disrupted by compound shifts, whereas DiffCrossGait better preserves cross-modal identity consistency through trajectory-level regularization.

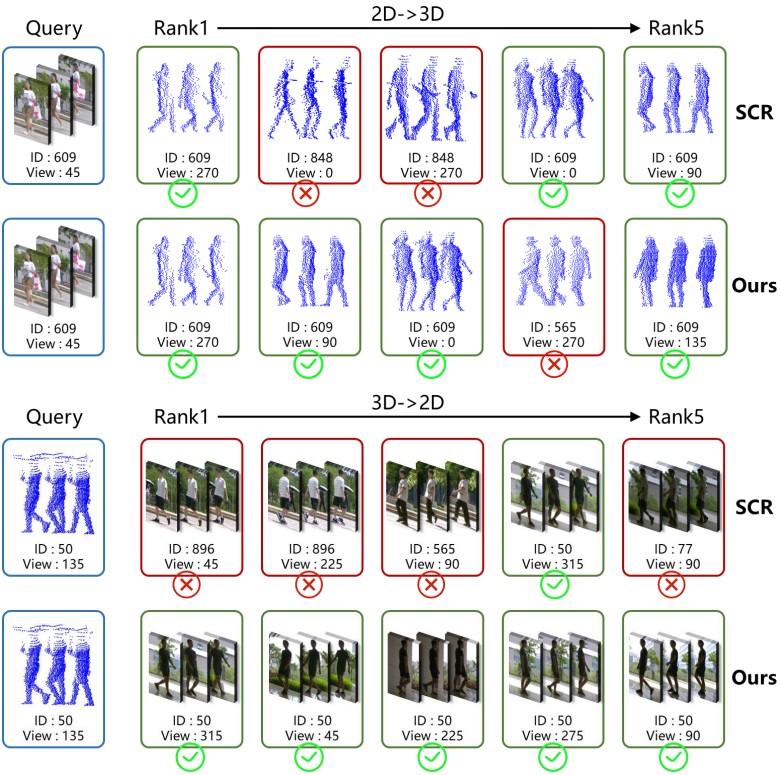

*Figure 6.* Representative retrieval examples from the challenging subset.

## A.6. Failure Cases

We also visualize representative failure cases in Fig. 7. The dominant failure pattern is extreme information scarcity: when night-time degradation and severe geometric distortion occur simultaneously, both modalities may lose identity-related structural anchors. In such cases, the denoising process lacks sufficient residual evidence to recover a shared clean latent state, causing cross-modal trajectory alignment to fail.

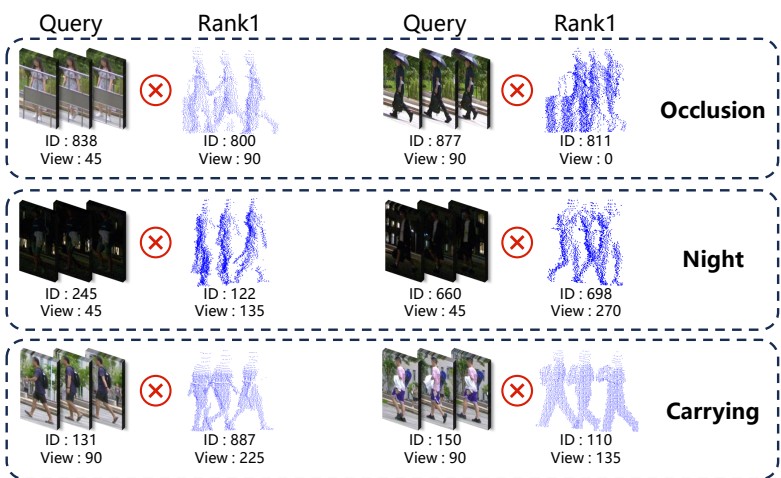

*Figure 7.* Representative failure cases. DiffCrossGait may fail when both visual and geometric structural cues are severely degraded, such as night-time degradation, occlusion, or severe carrying conditions.

