# OpenReview forum: "DiffCrossGait: Trajectory-Level Alignment for 2D-3D Cross-Modal Gait Recognition via Latent Diffusion"
_ICML.cc/2026/Conference — ICML 2026 regular_

### Official Review · Reviewer_nb4f · 2026-02-25

**Soundness:** 3
**Presentation:** 4
**Significance:** 3
**Originality:** 4
**Overall Recommendation:** 5
**Confidence:** 4

**Summary:**

DiffCrossGait proposes a trajectory-level alignment framework for cross-modal 2D–3D gait recognition. Instead of aligning only final embeddings, it enforces dynamic process alignment by driving both modalities with shared Gaussian noise in a unified latent diffusion process. A Tri-Phase Alignment Strategy constrains identity semantics, denoising dynamics, and cross-modal structural recoverability across different noise regimes. Diffusion is used only during training, preserving inference efficiency while achieving state-of-the-art results on SUSTech1K and FreeGait.

**Compliance With Llm Reviewing Policy:**

Affirmed.

**Final Justification:**

I raised my score to 5 since authors have addressed my concerns.

**Key Questions For Authors:**

1. Table 7 compares the relative training cost across different U-Net configurations. However, how much additional training cost does the method incur when diffusion is used as auxiliary supervision throughout training, compared to training without it?
2. The performance and alignment behavior appear sensitive to denoiser capacity (Table 7). Could the authors provide the exact U-Net configuration (number of layers, channel sizes, attention blocks, parameter count), and clarify whether it is convolutional, transformer-based, or hybrid?
3. What values have you chosen for $\rho_a$ and $\rho_b$? How do you pick the values, and how sensitive the training is to the values?

**Limitations:**

The paper includes an Impact Statement on page 9 stating, “There are many potential societal consequences of our work, none which we feel must be specifically highlighted here.” This feels insufficient.

Gait recognition is a remote biometric. It allows identification at a distance and does not require a person’s awareness or cooperation. Because of this, the potential for misuse is real. The authors should acknowledge the risks of large scale surveillance, tracking individuals without consent, and possible deployment by authoritarian or overly intrusive institutions. Even a short but thoughtful discussion of these concerns would show that the authors have reflected on the broader implications of strengthening cross modal identification systems.

The paper also does not present failure cases. As a reader, it is hard to understand when DiffCrossGait does not work well or when trajectory level alignment fails to provide an advantage. Showing examples where the method struggles, such as under extreme occlusion or severe sensor noise, would make the evaluation more transparent. Discussing limitations openly would not weaken the contribution. It would make the paper more credible and balanced.

**Strengths And Weaknesses:**

**Strengths:**
- Unlike other metric-based approaches, where alignment is enforced between 2D and 3D features only at the final embedding layer, this method embeds alignment directly within a shared latent space of a diffusion process. This is done by adding identical Gaussian noise to each modality each each timestep. thus ensuring that they are constrained to progress through the denoising process under the influence of a common noise process.
- The framework partitions diffusion time into small, medium, and large noise intervals and imposes different supervision in each regime. At low noise, identity anchoring preserves discriminative semantics and prevents drift; at medium noise, the model aligns noise prediction and bottleneck states to synchronize denoising dynamics; at high noise, cross-modal conditioning enforces structural recoverability under weak signal. This noise-aware curriculum is well-motivated: different timesteps correspond to different semantic resolutions, and the model leverages this to progressively enforce identity consistency, trajectory coupling, and manifold overlap.
- Alignment is enforced not only at reconstructed latents but also at the predicted noise level and at the bottleneck bridging state within the U-Net. The shared noise prediction loss ensures both modalities interpret the same perturbation consistently; the bottleneck state alignment directly constrains intermediate representations where modality-specific artifacts might otherwise diverge; reconstruction terms prevent degenerate collapse. This layered supervision makes the alignment structurally grounded rather than superficially enforced at the output layer, strengthening cross-modal invariance in a controllable manner.
- Diffusion is strictly a training-time auxiliary objective, and the entire diffusion branch is discarded at inference. As shown in the efficiency comparison, the method maintains nearly identical parameter count, FLOPs, and latency to the baseline backbone while achieving substantial accuracy gains.
- On SUSTech1K and FreeGait, the method achieves state-of-the-art results in both 2D→3D and 3D→2D retrieval settings, with particularly strong improvements under challenging covariates such as clothing changes, carrying conditions, and illumination variation.

**Weaknesses:**
- At line 323, the paper attributes the performance improvements to aligning denoising dynamics rather than endpoint alignment. While this interpretation is intuitive, it remains somewhat unclear whether the gains are primarily due to trajectory alignment itself or to the additional training constraints introduced by the diffusion framework. Although Table 6 demonstrates that the multi-phase objectives significantly improve performance, it does not fully isolate trajectory alignment as the main contributing factor. To further support this claim, it could be helpful to include diagnostics that directly quantify cross-modal alignment throughout the diffusion process. For instance, measuring the distance  || z_t^{2D} - Z_t^{3D} || across diffusion steps and comparing it between a baseline model and the full tri-phase alignment setting could provide insight into how alignment evolves. Such analysis could examine whether (i) cross-modal distances are consistently reduced when tri-phase alignment is applied, and (ii) the trajectories converge more tightly over time. This would help clarify the mechanism behind the observed improvements.
- The axial attention used in the denoising architecture is motivated as a mechanism to reduce overfitting to local textures; however, it introduces several potential limitations : (I) the assumption of a columnar structure may not hold under varying camera viewpoints and walking directions. While the vertical axis may align with the human body in sagittal views, this alignment is less meaningful for oblique or frontal viewpoints. (ii) important gait cues such as arm swing asymmetry or foot strike sharpness involve cross-axis interactions that may not be fully captured by axial attention. (iii) since LiDAR data is inherently irregular, imposing an axial bias through grid-based processing may introduce an artificial structural prior. Although Table 5 shows that adding axial attention improves overall performance, it remains unclear whether this improvement is consistent across viewpoints. Given that SUSTech1K is annotated by viewpoint angle, it would be helpful to report performance with and without axial attention across viewpoint bins to assess whether the benefit generalizes uniformly.
- In addition to the provided t-SNE visualization, qualitative experiments showing in which scenarios baselines fail but DiffCrossGait succeeds can provide further insight into the types of covariate shifts where trajectory alignment is most beneficial and help illustrate the practical advantages of the proposed approach.

---

> ### Author Rebuttal · Authors · 2026-03-30
>
> We thank the reviewer for the valuable feedback. Responses to specific comments are as follows:
>
> **(1) Trajectory Alignment Mechanism.**
>
> While measuring the distance between raw latents || z_t^{2D} - z_t^{3D} || is an intuitive starting point, the shared injected noise naturally causes this raw distance to shrink to zero as t→T, which partially obscure the network's actual learned alignment.
> Therefore, a more revealing diagnostic is the Euclidean gap between the predicted clean states:
>
> $\|\|\hat{z}_{0,t}^{2d}$
>
> $-$
>  $\hat{z}_{0,t}^{3d}\|\|_2$.
>
> Since a larger timplies heavier noise and weaker original signals, decoding the identical clean state from two different modalities becomes inherently harder. Thus, it is expected that this gap naturally increases as t grows; the key to robust alignment is restricting this divergence. On SUSTech1K cases, we observed the following gaps in post-hoc analysis:
>
> Baseline (No Diffusion): Evaluated on static final embeddings (used as an endpoint reference), yielding a gap of 0.23.
>
> DiffCrossGait (Ours): At the clean endpoint, our gap drops to 0.12, showing stronger endpoint alignment. Across the generative trajectory (t=10→50→90), our gap remains tightly controlled and coupled: 0.18 →0.31 →0.63.
>
> Phase 1-only Variant (Generative Baseline): Since standard baselines lack a diffusion trajectory (t>0), we benchmark against a Phase-1-only variant. Without mid/high-noise constraints, its cross-modal gap wildly diverges (0.34 at t=10→1.74at t=90), proving our multi-phase curriculum is essential to actively synchronize trajectories under severe noise.
>
> This diagnostic explicitly clarifies our mechanism: the performance gains stem from our multi-phase curriculum actively synchronizing the trajectories and preventing modal divergence under severe noise. Details in:
> https://anonymous.4open.science/r/ICML2026_Rebuttal-422F/Table1.md
>
>
> **(2) Axial Attention & Viewpoint Robustness.**
>
> We appreciate the reviewer's deep thoughts on the structural prior. However, because SUSTech1K is captured from ground-level cameras, the viewpoint variations are strictly horizontal (azimuth).
> Because humans stand upright, the body's topological order (head to toe) consistently spans the vertical spatial dimension (H) across all horizontal angles, whether frontal (0°) or profile (90°). Consequently, computing self-attention along the H-axis effectively captures this global anatomical structure regardless of the walking direction. This is validated by consistent improvements across all viewpoints (e.g., frontal 0°: 54.9→55.9; oblique 45°: 58.2→59.8; profile 90°: 66.1→68.4).
>
> Limitations & Failure Cases: We acknowledge that our columnar assumption breaks down under extreme pitch angles (e.g., top-down UAV surveillance), where the body's projection severely compresses and no longer aligns with the vertical H-axis, which has been added to our revised limitations.
>
> **(3) Qualitative Comparison/Failure Cases.**
>
> Comparison: (Example: https://anonymous.4open.science/r/ICML2026_Rebuttal-422F/Fig1.png) On a manually selected challenging subset, our method achieves 29/40 Top-1 retrievals versus 17/40 for SCR (fail under compound shifts (e.g., bag/unbrella)).
>
> Failures: (Example: https://anonymous.4open.science/r/ICML2026_Rebuttal-422F/Fig2.png) Analysis: Our method struggles when severe visual degradation (e.g., night) and geometric distortion (e.g., heavy carrying) occur simultaneously. When structural priors in \textit{both} modalities are severely obliterated, the diffusion process lacks sufficient residual anchors to reconstruct a shared clean latent state ($\hat{z}_0$), causing the cross-modal trajectory alignment to ultimately collapse.
>
> **(4) Config & Cost.**
>
> DiffCrossGait employs a 2.3M-param hybrid U-Net (scales: 16x16↔4x4, dim: 512→256) with 4 ResBlocks and 1 Axial-Attention bottleneck. Training overhead is +31.8% (11.2h vs 8.5h on RTX3090). Crucially, the diffusion branch is discarded during inference, ensuring near-zero extra computational cost.
>
> **(5) Hyperparameters $\rho_a, \rho_b$.**
>
> We will correct $(\tau_a, \tau_b)$ to $(\rho_a, \rho_b)$ (In Appendix) for consistency.
>
> Rationale: $\rho_a=0.95$ (minimal noise) anchors identity; $\rho_b=0.50$ (median SNR) balances Phase 2 (dynamics) and Phase 3 (structure).
>
> Sensitivity (SUSTech1K 2D$\rightarrow$3D Rank-1):
> Varying $\rho_a$ (given $\rho_b=0.50$): 0.90→58.3%, 0.95→58.7% (Ours), 0.98→58.5%;
> Varying $\rho_b$ (given $\rho_a=0.95$): 0.40→58.1%, 0.50→58.7% (Ours), 0.60→58.4%;
> The results confirm that while the model peaks around our chosen semantic thresholds, it remains highly stable (fluctuations <0.6%) across a wide range of schedule partitions.
>
> **(6) Broader Impact.**
>
> The revision explicitly addresses remote biometric risks (mass surveillance, non-consensual tracking) and emphasizes the necessity for strict legal governance.

---

> > ### Author Rebuttal · Reviewer_nb4f · 2026-04-02
> >
> > The authors have addressed my concerns. I'll raise my score accordingly.

---

> > > ### Author Response · Authors · 2026-04-05
> > >
> > > Thank you very much for your thoughtful follow-up and for letting us know that our rebuttal has addressed your concerns. We sincerely appreciate your careful reading of the paper and your constructive feedback on the trajectory alignment mechanism, denoising architecture, qualitative analysis, computational cost, and broader impact.
> > >
> > > We are especially grateful for your helpful suggestions on diagnostic analysis, viewpoint robustness, failure cases, and ethical considerations. In the camera-ready version, we will further strengthen these parts to make the paper more transparent, better motivated, and more complete.
> > >
> > > Thank you again for your time, valuable feedback, and encouraging support.

---

### Official Review · Reviewer_XKi1 · 2026-03-03

**Soundness:** 3
**Presentation:** 3
**Significance:** 2
**Originality:** 2
**Overall Recommendation:** 4
**Confidence:** 3

**Summary:**

The manuscript presents a diffusion-regularized framework, DiffCrossGait, for cross-modal gait recognition, addressing the challenging task of matching 2D silhouettes to 3D point clouds. In addition to conventional endpoint embedding alignment, the proposed method introduces trajectory-level alignment by driving both modalities with shared Gaussian noise within a unified latent diffusion process. The framework further stabilizes the training procedure through a Tri-Phase Alignment Strategy that includes identity anchoring, dynamics consistency, and cross-modal structural recoverability, and this strategy is applied across different noise regimes. Extensive experimental evaluations on the SUSTech1K and FreeGait benchmarks demonstrate that DiffCrossGait achieves state-of-the-art performance, which confirms the effectiveness and novelty of the proposed framework.

**Compliance With Llm Reviewing Policy:**

Affirmed.

**Final Justification:**

The response reinforced my prior assessment of the manuscript.

**Key Questions For Authors:**

see weaknesses

**Limitations:**

yes

**Strengths And Weaknesses:**

Strengths
The manuscript presents a clear idea by moving from static embedding alignment to trajectory-level alignment. The overall framework is coherent and technically sound, and the experimental results consistently demonstrate improvements over competitive baselines. In particular, on the FreeGait dataset, it achieves a notably larger improvement of 18.4% rank-1 acc, further supporting the effectiveness and novelty of the proposed approach. Overall, the study addresses a practically relevant problem and offers a structured solution supported by solid empirical validation.

Weaknesses
Problem formulation could be better motivated. The manuscript does not clearly justify why explicit 2D–3D cross-modal matching is required, rather than first mapping 3D representations into the 2D domain and performing comparison there. Without discussing the limitations of such projection-based approaches, the necessity of trajectory-level cross-modal alignment remains somewhat under-motivated.
Noise schedule and phase boundary sensitivity are not fully discussed. The tri-phase alignment strategy depends on the noise schedule and the thresholds ρ_a and ρ_b used to partition the diffusion timeline. While the paper states that the partition is derived from the schedule, it is not entirely clear how sensitive the performance is to these hyperparameters. For example, would slightly different threshold values significantly affect alignment behavior? Is the improvement mainly due to the phased design itself, or to a specific choice of boundaries? A brief sensitivity analysis or discussion would help clarify whether the method is robust to such design choices.

---

> ### Author Rebuttal · Authors · 2026-03-30
>
> We sincerely thank the reviewer for raising these insightful questions.
>
> **Q1: Necessity of Explicit 2D-3D Cross-Modal Alignment vs. 3D-to-2D Projection**
>
> While mapping 3D to 2D intuitively bridges the modality gap, explicit cross-modal alignment (e.g., Cross-modal Gait Recognition, VI-ReID) is necessary to preserve **modality-invariant, fine-grained discriminative representations.** Cross-modal gait recognition demands a **delicate input-level balance**: minimizing the modality gap for feasible alignment while maximizing fine-grained information retention. We clarify our choices across three representation levels:
>
> **1. Why use LiDAR Range-View instead of Artificial 2D Projections?**
> To ensure a strictly fair comparison, our framework strictly follows the established input paradigm of recent cross-modal gait benchmarks (e.g., CrossGait, CL-Gait), which utilize 2D silhouettes and LiDAR range-views. It is crucial to note that a LiDAR range-view is **not** an artificial 3D-to-2D depth projection; it is a **sensor-native modality** that naturally preserves the dense spherical geometry, exact depth cues, and fine-grained textures captured by the LiDAR.
>
> In contrast, artificially projecting 3D data into a flattened 2D space (e.g., depth maps or pseudo-silhouettes) inevitably causes an irreversible loss of these fine-grained identity details. While projection makes the two modalities structurally more similar and superficially easier to align, it heavily discards the complementary discriminative signals essential for robust identification under complex covariates.
>
> **2. Why not use Raw 3D Point Clouds?**
> One might ask: if preserving spatial geometry is critical, why not use raw 3D point clouds directly? In our preliminary experiments, we found that the highly unstructured and sparse nature of raw point clouds introduces an excessively massive modality gap when paired with 2D silhouettes; the Rank-1 accuracy of the baseline severely dropped to 10.1% (2D→3D) and 11.9% (3D→2D), leading to alignment collapse. The range-view serves as the optimal balance point: it maintains the grid-like structural compatibility needed for stable cross-modal feature extraction while retaining high-fidelity geometric priors.
>
> **3. Quantitative Evidence.**
> To explicitly quantify how artificial 2D projection degrades discriminative power, we designed a controlled ablation. We forcefully converted the native LiDAR range-view into a 2D pseudo-silhouette format. Evaluated under the same retrieval protocol, the Rank-1 accuracy of the baseline severely dropped to 27.6% (2D→3D) and 28.2% (3D→2D). This sharp decline explicitly demonstrates that prematurely forcing 3D-oriented data into a 2D contour space destroys the network's ability to extract identity-discriminative features.
>
> **Conclusion:** Because we choose to maintain the native, geometry-rich range-view to preserve identity cues, the latent embedding gap naturally remains too large for traditional static endpoint alignment to handle effectively. Consequently, this fundamental trade-off directly motivates our **trajectory-level alignment.** By employing a shared-noise latent diffusion process, DiffCrossGait effectively aligns the semantic evolution without needing to destructively project the inputs beforehand. We will explicitly incorporate this discussion in the revised manuscript.
>
> **Q2: Sensitivity to noise schedule and phase boundaries ($ρ_a$, $ρ_b$).**
>
> **1. The Source of Improvement.**
>
> As ablated in Table 6, the phased curriculum design itself is the primary driver of the improvement. Crucially, as defined by the schedule-derived partition in Section 3.3, these chosen thresholds are not arbitrary but naturally align with the semantic properties of the diffusion process. The threshold $ρ_a$=0.95(minimal noise) securely anchors the initial identity, while $ρ_b$=0.50(median SNR) serves as a balanced watershed between dynamic consistency (Phase 2) and structural recovery (Phase 3).
>
> **2. Sensitivity Analysis.**
>
> We evaluated different threshold combinations on the SUSTech1K dataset (2D→3D Rank-1), with performance fluctuations ≤0.6%:
>
> | Configuration | Value | Result |
> | :- | :-: | :-: |
> | **Varying $\rho_a$** | 0.90 | 58.3 |
> | *(with $\rho_b = 0.50$)* | **0.95 (Ours)** | **58.7** |
> | | 0.98 | 58.5 |
> | **Varying $\rho_b$** | 0.40 | 58.1 |
> | *(with $\rho_a = 0.95$)* | **0.50 (Ours)** | **58.7** |
> | | 0.60 | 58.4 |
>
> Thus, while the phased design is critical, exact boundary values do not cause catastrophic shifts, proving robustness. Furthermore, testing different diffusion lengths (T = 50, 100, 200) yielded highly stable Rank-1 accuracies (58.2%, 58.7%, 58.6%), validating insensitivity to schedule variations.
>
> _(We appreciate the reviewer's careful reading; we will ensure consistent notation—using $ρ_a$,$ρ_b$ instead of $τ_a$,$τ_b$ —in the revised manuscript.)_

---

> > ### Author Rebuttal · Reviewer_XKi1 · 2026-04-02
> >
> > My concerns have been adequately addressed.

---

> > > ### Author Response · Authors · 2026-04-05
> > >
> > > Thank you very much for your follow-up and for confirming that our rebuttal has adequately addressed your concerns. We sincerely appreciate your careful reading of our work and your thoughtful questions regarding the motivation for explicit cross-modal alignment and the robustness of the tri-phase design.
> > >
> > > We are glad that our clarification helped resolve these issues. In the camera-ready version, we will further strengthen the discussion of the motivation behind explicit 2D–3D alignment, and clarify the robustness of the method with respect to the noise schedule and phase boundaries.
> > >
> > > Thank you again for your time and valuable feedback.

---

### Official Review · Reviewer_A9HD · 2026-03-06

**Soundness:** 2
**Presentation:** 3
**Significance:** 2
**Originality:** 3
**Overall Recommendation:** 4
**Confidence:** 5

**Summary:**

This submission is focused on the task of gait recognition across different modalities, i.e., 2D silhouettes and 3D point clouds. The main idea is to align the two types of feature representations using the diffusion denoising method. The main contributions of the submission include 1) a new approach, i.e., DiffCrossGait, for cross-modal gait recognition by trajectory-level alignment using diffusion, 2) a Tri-Phase Alignment Strategy to learn modality-invariant representations, and 3) a group of elaborately designed objectives for modeling training. Experiments on two public benchmarks show superior performance of the proposed framework.

**Compliance With Llm Reviewing Policy:**

Affirmed.

**Final Justification:**

In the response of the authors, my main concerns in the original review are addressed. So I upgrade the score to weak accept.

**Key Questions For Authors:**

- Q1. Despite the significance of the proposed framework for cross-modal gait recognition, can it also be applied to other multi-modal tasks like cross-modal retrieval, cross-modal face recognition, cross-modal person re-identification, etc?
- Q2. How are the hyperparameters in Equation (18) selected or balanced?
- Q3. How is the Base in Table 6 implemented? Why does phase 1 obtain the most significant improvement, while the other two phases are weak?

**Limitations:**

The authors provide a brief discussion about the potential impact, but neglect the discussion about the limitations of the proposed framework. I suggest the authors discuss its limitations with the bad cases in the experiments.

**Strengths And Weaknesses:**

- S1. The manuscript is well-written and easy to follow. The figures and tables are instructive.
- S2. The idea of using diffusion denoising for cross-modal alignment in gait recognition is interesting and novel. The proposed framework and strategy are technically sound.
- S3. The experimental results on public benchmarks are good to validate the effectiveness of the proposed framework.

- W1. Although the idea of using diffusion denoising to align the two modalities of gait sequences is interesting and novel, the overall learning process and the structure of the framework are very complex. This may hinder its application to other datasets or tasks and narrow its impact in the community of machine learning.
- W2. From the ablation study in Table 6, it seems that phase 1 is the most significant part in the strategy, while the other two seem weaker than phase 1.

---

> ### Author Rebuttal · Authors · 2026-03-30
>
> We thank the Reviewer for the insightful feedback and address the concerns with detailed evidence below:
>
> **1. On Framework Complexity and Generalizability (W1 & Q1)**
>
> We appreciate the chance to note: DiffCrossGait is designed as a modular, training-only regularizer, not a complex ad-hoc architecture:
>
> - Inference Efficiency: The diffusion branch is an auxiliary objective used exclusively only during training and is **discarded at inference**. As validated in Table 4, DiffCrossGait maintains nearly the same inference footprint as the baseline (14.0M vs. 13.7M params; 6.2ms vs. 5.9ms latency) while delivering a significant +9.1% Rank-1 gain.
>
> - Plug-and-play Generalization: The framework is backbone-independent. Its core principle—trajectory-level alignment via shared noise—is modality-agnostic. To apply it to other tasks (e.g., cross-modal Re-ID or Face Recognition), one only needs to append the diffusion objective to the latent space without altering the task-specific backbone (e.g., ResNet, ViT).
>
> - Empirical Impact: As a proof-of-concept, we applied our objective on SYSU-MM01 dataset (all-search, single-shot):
>
> DEEN (CVPR 2023): Rank-1/mAP increased from 74.70/71.80% to 75.19/72.01%.
>
> IDKL (CVPR 2024): Rank-1/mAP increased from 81.42/79.85% to 81.91/80.11%.
>
> By transitioning from static endpoint matching to dynamic trajectory synchronization, we offer a principled alternative to traditional metric learning, with broad potential for heterogeneous data alignment in the ML community.
>
> **2. On the Base Model and Tri-Phase Synergy (W2 & Q3)**
>
> The Base model (Table 6, the same as baseline in Table 4) uses only standard $L_{disc}$ losses, with all diffusion disabled.
>
> We agree Phase 1 is a crucial foundational anchor. By applying identity supervision at small noise, it prevents semantic drift in the generative branch. However, Phases 2 and 3 are not weak; they tackle the most challenging structural domain gaps. They enforce modality-invariant recovery under extreme covariates (e.g., "Clothing", "Umbrella"), which is critical for achieving our SOTA 63.8% performance.
>
> Furthermore, even if Phase 1 is removed (Exp 4: 61.8%), the model remains robust because our decoupled architecture still preserves fundamental discriminative supervision in the main branch. Ultimately, Phase 1 safely bridges the discriminative and generative spaces, but it is the trajectory constraints of Phases 2 & 3 that fundamentally resolve the 2D-3D structural discrepancies.
>
> **3. On Hyperparameter Selection and Stability (Q2)**
>
> Our parameter selection follows a hierarchical prioritization logic: we set $\lambda_1=1.0$ (Primary task) as the anchor, assigned a moderate weight $\lambda_2=0.5$ (Identity Anchoring) to bridge the two branches, and kept the trajectory/reconstruction terms ($\lambda_{3,4,5}$) relatively small to provide auxiliary guidance.
> To address stability concerns, we conducted a sensitivity analysis on SUSTech1K (Rank-1, 3D$\rightarrow$2D). As shown below, our method is robust across a range of weight combinations:
>
> | Configuration ($\lambda_2, \lambda_{3,4}, \lambda_5$) | Rank-1 (%) | Rank-5 (%) |
> | :-| :-: | :-: |
> | Case A: (0.1, 0.1, 0.05)                              |    62.1    |    82.2    |
> | Case B: (1.0, 0.1, 0.05)                              |    63.3    |    82.9    |
> | Case C: (0.5, 0.05, 0.05)                             |    62.9    |    82.5    |
> | Case D: (0.5, 0.2, 0.1)                               |    63.1    |    82.7    |
> | **Ours: (0.5, 0.1, 0.05)**                        |  **63.8**  |  **83.4**  |
>
> The exact same set of hyperparameters are used for both SUSTech1K and FreeGait datasets.
>
> **4. On Limitations and Diagnostic Analysis**
>
> - Comparison visualization: Compared to the SOTA method SCR, our method achieves 29/40 Top-1 retrievals (vs. 17/40 for SCR) on a manually curated hard subset, indicating its superior robustness under disrupted contours. (Example : https://anonymous.4open.science/r/ICML2026_Rebuttal-422F/Fig1.png),.
>
> - Failure visualization: We identify the primary limitation as Extreme Information Scarcity.  This occurs when severe visual degradation (e.g., night) and heavy geometric distortion (e.g., severe occlusion) happen simultaneously. In these edge cases, both modalities lose essential structural priors. The latent diffusion process lacks sufficient residual anchors to reconstruct a shared clean state $\hat{z}_0$, causing the cross-modal trajectory alignment to ultimately collapse. (Example: https://anonymous.4open.science/r/ICML2026_Rebuttal-422F/Fig2.png)
>
> - Broader Impact: Following your advice, we will expand the discussion on ethical deployment to ensure that denoising focuses on identity-invariant structural features without leaking sensitive gait textures.

---

> > ### Author Rebuttal · Reviewer_A9HD · 2026-04-02
> >
> > My concerns in the original review are solved. So I will update my rating.

---

> > > ### Author Response · Authors · 2026-04-05
> > >
> > > Thank you very much for your follow-up and for confirming that our rebuttal has addressed your concerns. We sincerely appreciate your careful reading of our work and your thoughtful suggestions regarding the framework complexity, the role of each phase, hyperparameter selection, and the discussion of limitations.
> > >
> > > We are grateful that our clarification helped resolve these issues. In the camera-ready version, we will further refine the relevant sections to make the implementation details, design motivation, and limitations of the method even clearer.
> > >
> > > Thank you again for your valuable feedback and support.

---

### Official Review · Reviewer_UGRZ · 2026-03-25

**Soundness:** 2
**Presentation:** 3
**Significance:** 3
**Originality:** 3
**Overall Recommendation:** 4
**Confidence:** 3

**Summary:**

This paper presents DiffCrossGait, a diffusion-regularized framework for cross-modal 2D–3D gait recognition. Instead of aligning only the final embeddings of 2D silhouettes and 3D point clouds, the method reformulates cross-modal matching as a trajectory-level alignment problem in a shared latent diffusion space. The core idea is to inject shared Gaussian noise into both modalities and encourage them to follow compatible denoising dynamics throughout the generative process. To support this, the paper introduces a Tri-Phase Alignment Strategy with identity anchoring, dynamics consistency, and structural recoverability constraints. The framework uses diffusion only during training, so it avoids extra inference cost while improving cross-modal recognition performance. Extensive comparisons against the existing approaches demonstrate the superiority of the proposed algorithm.

**Compliance With Llm Reviewing Policy:**

Affirmed.

**Key Questions For Authors:**

None.

**Limitations:**

yes

**Strengths And Weaknesses:**

Strengths:
1. The paper addresses a meaningful and timely problem, since the modality gap between 2D silhouettes and 3D point clouds remains a central challenge in cross-modal gait recognition.
2. The main idea is conceptually appealing: instead of only matching the final features, the method aligns the process of representation evolution, which feels well motivated for gait because identity is inherently tied to motion dynamics.
3. The proposed Tri-Phase Alignment Strategy is nicely structured, and the three components, identity anchoring, dynamics consistency, and structural recoverability, work together in a coherent way rather than feeling like unrelated losses.
4. I also like that the diffusion module is used only during training, because this makes the method easier to justify in practice and avoids the common concern that generative modeling will slow inference.
5. Detailed ablation studies are provided to show the impact of each component.

Weaknesses:

1. The method appears to assume that the two modalities should share compatible diffusion dynamics in a common latent space, but this assumption may be stronger than necessary and is not yet fully justified from a representation-learning standpoint. Since 2D silhouettes and 3D point clouds differ not only in noise level but also in information content and geometric observability, full trajectory compatibility may not always be theoretically well aligned with optimal identity preservation.

---

> ### Author Rebuttal · Authors · 2026-03-30
>
> We sincerely thank the reviewer for the positive feedback and for recognizing the conceptual appeal of our trajectory-level alignment and the efficiency of our zero-inference-cost design.
>
> You raised an exceptionally insightful point regarding the inherent differences in information content and geometric observability between 2D and 3D modalities. We completely agree that naively assuming full trajectory equivalence across highly heterogeneous modalities could conflict with optimal identity preservation. We would like to clarify that DiffCrossGait does not posit strict full-modality equivalence; rather, it enforces a weaker, adaptive constraint designed specifically to bridge this gap without distorting identity cues:
>
> **1. Standardized Input Representations:** First, to mitigate the initial geometric gap, our alignment is not imposed on unconstrained raw sensory inputs (e.g., raw RGB videos vs. raw unordered point clouds). Following standard cross-modal gait protocols, we utilize silhouettes for the camera modality and range-view representations for LiDAR. This provides a more comparable, gait-centric structural baseline before any diffusion constraint is applied.
>
> **2. Decoupled Architecture for Optimal Identity Preservation:** Our intended assumption is that paired samples should admit compatible denoising dynamics within an identity-relevant shared subspace, while modality-specific nuances remain outside that subspace. This is precisely why we designed the Decoupled-Head architecture (Sec. 3.1). The shared-noise diffusion regularization is exclusively applied to the generative latent ($z_{gen}$), while the discriminative latent ($z_{disc}$) remains fully dedicated to metric learning. This ensures the generative objective transfers modality-invariant structure without stifling fine-grained identity cues. As shown in Table 5, removing these decoupled heads directly degrades performance.
>
> **3. Adaptive, Non-Rigid Constraints (Tri-Phase Strategy):** Rather than imposing a uniform, rigid constraint across the entire trajectory, our alignment is adaptive. As validated by our ablations (Table 6), the Tri-Phase strategy explicitly accommodates the modalities' differing information capacities: Phase 1 preserves identity semantics under low noise, Phase 2 synchronizes general dynamics, and Phase 3 specifically tackles the structural recoverability gap when the signal is weak. Specifically, Phase 3 (CrossCond) forces the modalities to establish a reciprocal structural recovery mechanism. By using the 3D prior to drive 2D generation (and vice versa) under large noise, the network explicitly learns to bridge the observability gap, rather than forcing an impossible one-to-one geometric mapping.
>
> **4. Theoretical Alignment (Representation Learning):**
> From a representation-learning standpoint, matching the denoising trajectories driven by shared noise is mathematically equivalent to bounding the Fisher Divergence between the two latent distributions (detailed via Score Matching in Appendix A.3). This trajectory-level shared-noise training aligns the score/denoising fields rather than just the endpoint means, acting as a structural regularizer for cross-modal manifold alignment.
>
> We deeply appreciate this constructive critique. In the camera-ready version, we will revise Sections 3.2–3.3 and the Limitations paragraph to explicitly state that DiffCrossGait uses shared-noise diffusion strictly as a training-time structural regularizer for an identity-relevant subspace, rather than assuming full modality equivalence.

---

> > ### Author Rebuttal · Reviewer_UGRZ · 2026-04-03
> >
> > My concerns have been adequately addressed.

---

> > > ### Author Response · Authors · 2026-04-05
> > >
> > > Thank you very much for your follow-up and for confirming that our rebuttal has adequately addressed your concerns. We sincerely appreciate your careful reading of our work and your thoughtful feedback throughout the review process.
> > >
> > > We are pleased that our clarification helped resolve the concern regarding cross-modal diffusion dynamics. In the camera-ready version, we will further refine the relevant discussion to make the intended assumptions, applicability, and limitations of the method more explicit.
> > >
> > > Thank you again for your time and consideration.

---

### Decision · Program_Chairs · 2026-04-30

**Decision:**

Accept (regular)

**Comment:**

This paper proposes DiffCrossGait, a framework that reformulates cross-modal 2D–3D gait recognition as a trajectory-level alignment problem within a shared latent diffusion space. All the reviewers reached consensus that this is a technically solid paper with a novel and well-motivated contribution. The authors' rebuttal comprehensively addressed all raised concerns, and the paper advances the state of the art with a principled approach.

The AC is happy to accept this paper to ICML. Congratulations!  Meanwhile, please keep in mind that the authors should incorporate the promised revisions in the camera-ready version, particularly:

- (i) clarifying the shared-dynamics assumption and its scope,
-  (ii) adding the failure case analysis and broader impact discussion,
- and (iii) including the sensitivity and generalizability experiments from the rebuttal.